# The blood-brain barrier studied *in vitro* across species

**Maj Schneider Thomsen**, **Nanna Humle**, **Eva Hede**, **Torben Moos**,
**Annette Burkhart**‡*, **Louiza Bohn Thomsen**‡

Department of Health Science and Technology, Neurobiology Research and Drug Delivery, Aalborg University, Aalborg, Denmark

☯ These authors contributed equally to this work.
‡ AB and LBT also contributed equally to this work.
* abl@hst.aau.dk

**Data Availability Statement:** All relevant data are within the manuscript.

**Funding:** The work was supported by grants from Fonden til Lægevidenskabens Fremme (https://www.apmollerfonde.dk/ansoegning/laegefonden/)

## Abstract

The blood-brain barrier (BBB) is formed by brain capillary endothelial cells (BECs) supported by pericytes and astrocytes. The BBB maintains homeostasis and protects the brain against toxic substances circulating in the blood, meaning that only a few drugs can pass the BBB. Thus, for drug screening, understanding cell interactions, and pathology, *in vitro* BBB models have been developed using BECs from various animal sources. When comparing models of different species, differences exist especially in regards to the transendothelial electrical resistance (TEER). Thus, we compared primary mice, rat, and porcine BECs (mBECs, rBECs, and pBECs) cultured in mono- and co-culture with astrocytes, to identify species-dependent differences that could explain the variations in TEER and aid to the selection of models for future BBB studies. The BBB models based on primary mBECs, rBECs, and pBECs were evaluated and compared in regards to major BBB characteristics. The barrier integrity was evaluated by the expression of tight junction proteins and measurements of TEER and apparent permeability (Papp). Additionally, the cell size, the functionality of the P-glycoprotein (P-gp) efflux transporter, and the expression of the transferrin receptor were evaluated and compared. Expression and organization of tight junction proteins were in all three species influenced by co-culturing, supporting the findings, that TEER increases after co-culturing with astrocytes. All models had functional polarised P-gp efflux transporters and expressed the transferrin receptor. The most interesting discovery was that even though the pBECs had higher TEER than rBECs and mBECs, the Papp did not show the same variation between species, which could be explained by a significantly larger cell size of pBECs. In conclusion, our results imply that the choice of species for a given BBB study should be defined from its purpose, instead of aiming to reach the highest TEER, as the models studied here revealed similar BBB properties.

## Introduction

The blood-brain barrier (BBB) is a protective barrier formed by non-fenestrated brain capillary endothelial cells (BECs), which are supported by pericytes embedded in the basement membrane and astrocytic endfeet [1]. Pericytes cover approximately 30% of brain capillaries, while

(MST, NH, AB, LBT), Kong Christian d. X Fond (http://kongehuset.dk/node/5556) (NH), Fhv. Dir. Leo Nielsen og Hustru Karen Margrethe Nielsens Legat for Lægevidenskabelig Grundforskning (https://www.njordlaw.com/da/om-njord/fonde-og-legater/fhv-dir-leo-nielsen-og-hustru-karen-margrethe-nielsens-legat-for-laegevidenskabelig-grundforskning/) (MST), Augustinus fonden (https://augustinusfonden.dk/) (AB, LBT), Danielsens fond (https://danielsensfond.dk/)(AB, LBT), the Lundbeck Foundation Research Initiative on Brain Barriers and Drug Delivery (Grant no. R155-2013-14113) (https://www.lundbeckfonden.com/) (TM). The funders had no role in study design, data collection and analysis, decision to publish, or preparation of the manuscript.

**Competing interests:** The authors have declared that no competing interests exist.

**Abbreviations:** *Actb*, β-actin; **A-L**, Abluminal to luminal luminal; **BBB**, Blood-brain barrier; **BCRP**, Breast cancer resistance protein; **BECs**, Brain capillary endothelial cells; **bFGF**, Basic fibroblast growth factor; **BSA**, Bovine serum albumin; **CLD**, Claudin; **CNS**, Central nervous system; **CTP-cAMP**, 8-(4-Chlorophenylthio)adenosine 3',5'-cyclic monophosphate sodium salt; **DMEM-F12**, Dulbecco's modified Eagle medium with nutrient mixture F12; **ER**, Efflux ratio; **FCS**, Fetal calf serum; **GFAP**, Glial fibrillary acidic protein; **GLUT1**, Glucose transporter 1; **hBECs**, Human brain endothelial cells; **HC**, Hydrocortisone; *Hprt1*, Hypoxanthine phosphoribosyltransferase 1; **JAMs**, Junctional adhesion molecules; **L-A**, Luminal to abluminal; **mBECs**, Mouse brain capillary endothelial cells; **OCLN**, Occludin; **Papp**, Apparent permeability; **pBECs**, Porcine brain capillary endothelial cells; **PBS**, Phosphate buffered saline; **P-gp**, P-glycoprotein; **rBECs**, Rat brain capillary endothelial cells; **RO**, RO-1724; **SD**, Standard deviation; **TEER**, Transendothelial electrical resistance; **TfR**, Transferrin receptor; **ZO**, Zonula occludens; **ZSQ**, Zosuquidar.

the astrocytic endfeet cover approximately 99% [2]. The BBB protects the vulnerable neurons against potentially harmful substances circulating in the blood by controlling the entrance of blood-borne substances into the brain [3]. However, the restrictive properties of the BBB become a major hurdle in the treatment of diseases affecting the central nervous system (CNS), as only a few drugs can pass the BBB and enter the CNS [4]. This has created a great interest in and a need for the development of *in vitro* BBB models that mimic the *in vivo* BBB.

The BECs are highly sophisticated cells characterized by high transendothelial electrical resistance (TEER), low permeability, low vesicular transport, and high expression of tight junction proteins, various transporters, and receptors important for maintaining the restrictive function of the BECs. The tight junction proteins claudin (CLDs), occludin (OCLN), and junctional adhesion molecules (JAMs) form the paracellular barrier and these are closely associated with the scaffolding and regulatory proteins zonula occludens (ZO)-1-3 [1, 5, 6]. The BECs also express efflux transporters of the ABC family such as breast cancer resistance proteins (BCRP) and P-glycoprotein (P-gp), which furthermore hinder brain delivery of small lipophilic drugs [7–9]. To ensure the supply of important nutrients to the brain, BECs also express several specific solute carriers like the glucose transporter 1 (GLUT1) and receptors such as the transferrin receptor 1 (TfR) [1]. The presence of pericytes and astrocytes supports the characteristics of the BECs through cell-cell interactions and communication through solute factors [10–14]. *In vitro* BBB models, should, therefore, mimic as many of these characteristics as possible and this have led to the creation of many different models, some more complex than others.

The simplest *in vitro* BBB models are created from immortalized cell lines. These models are easy and cheap to construct and highly suitable for high throughput screening of potential drug candidates. Unfortunately, these models do not mimic the complexity of the BBB *in vivo* [15, 16], why primary cells have become the preferred cell type, as these maintain many of the *in vivo* characteristics, like the expression of important tight junction proteins, solute carriers, receptors, efflux transporters, and they display high TEER and low permeability [15, 16]. The Transwell system, which is a widely used system in the construction of *in vitro* BBB models, allows for the formation of a polarized BEC layer with defined luminal and abluminal compartments, and it further provides co-culturing possibilities with pericytes and/or astrocytes. Co-culturing the BECs with pericytes and/or astrocytes in so-called co- and triple-culture models increases and maintains many of the *in vivo* characteristics of the BECs [13, 17–19]. Primary BECs, pericytes, and astrocytes have been isolated from several different species and used in the search for the optimal *in vitro* BBB model. The most extensively studied models utilize primary cells isolated from mice, rats, porcine, or bovine brains [15, 20–33].

Primary BECs isolated from different species all display several of the important *in vivo* characteristics, however when comparing *in vitro* BBB models constructed from primary cells of different species, one obvious difference has become evident. The TEER is highly variable across species, with the primary mBECs and rBECs exhibiting lower TEER values than pBECs and bovine BECs [20, 22, 34, 35]. This has created an ongoing discussion within the BBB community regarding the most favorable model, and whether the mice and rat models should be considered leaky and therefore unfavorable due to the lower TEER values obtained using these models, compared to those obtained with the porcine and bovine cells. Two methods are normally used to evaluate the BBB integrity *in vitro*, with the first being TEER and the second being the apparent permeability (Papp) of small hydrophilic molecules. Of the two, TEER is the most widely used, due to its easy and convenient use [36]. To our knowledge, no study to this date has investigated and compared the relationship between the Papp and TEER in different *in vitro* models constructed from BECs of different origin or have attempted to explain the significant differences observed in TEER among species.

The present study, therefore, compares three species-specific *in vitro* BBB models, to investigate the underlying reason for the major variation observed in TEER between species, but also to add to the selection of *in vitro* BBB models for future studies. The three species-specific *in vitro* BBB models studied are the mice, rat, and porcine models, as these are all well-characterized and highly used in our laboratory [22, 28, 31, 37–42]. We have previously characterized mice, rat, and porcine mono-, co-, and triple-cultures, showing that co-culturing the BECs with both astrocytes and pericytes do not further increase TEER significantly or cause significantly lower apparent permeability to small molecules compared to co-cultures with only astrocytes [22, 28, 31]. Thus, for the objective of the present study, we constructed the two most commonly used *in vitro* BBB models, namely the mono-culture and the non-contact co-culture model using primary mice, rat, and porcine BECs (mBECs, rBECs, pBECs) and primary astrocytes of the same origin. The primary BECs of the present study were all derived from the same isolation protocol with only minor modifications, making the models highly comparable. The models were compared with regards to major BECs characteristics, like the expression of tight junction proteins, efflux transport, expression of the TfR; a widely studied target for drug delivery to the brain [43], and most importantly, the barrier integrity.

## Materials and methods

All materials were obtained from Sigma-Aldrich unless otherwise stated.

### Isolation of mBECs, rBECs, and pBECs

Isolation of mBECs, rBECs, and pBECs has previously been described in [22, 28, 31]. The method was originally adapted from the isolation of rBECs and has later been slightly modified for application in mBECs and pBECs. The modifications were adopted in the first step of the purification procedure.

Handling of rats and mice took place in the animal facility at Aalborg University and was approved by the Danish National Council of Animal Welfare. The porcine brains were obtained from approximately six-month-old domestic pigs, donated by the local abattoir (Danish Crown, Sæby, Denmark). A single batch of mBECs requires 12–15, six to teen week old C57BL/6 mice brains, and 9–12, two-to-three weeks old Sprague-Dawley rat brains are needed for one batch of rBECs. Mice and rats were deeply anesthetized with isoflurane and sacrificed by decapitation. The forebrains were dissected under sterile conditions, and the meninges and visible white matter were removed. The porcine brains were transported on ice from the local abattoir to the laboratory facility. The meninges were removed and for a single batch of cells, 12 grams of grey matter was carefully dissected. From here on the protocols for isolation of BECs are identical across species.

The collected brain tissue was minced and digested in Dulbecco's modified Eagle medium with nutrient mixture F12 (DMEM-F12) (Life technologies) with collagenase type II (1mg/ml; Life Technologies) and DNase I (20μg/ml) at 37°C for 75 minutes in an incubating mini shaker (VWR). The digested brain material was pelleted by centrifugation and resuspended in 20% bovine serum albumin (BSA) (Europa Bioproducts) in DMEM and centrifuged at 1000 x *g* for 20 min. The pellet was further digested in DMEM-F12 containing collagenase-dispase (0.75mg/ml) and DNase I (7.5μg/ml) for 50 minutes at 37°C in the incubating mini shaker. The digested material was again pelleted by centrifugation and loaded on a 33% Percoll gradient to obtain microvessels. The microvessels were collected and washed twice in DMEM-F12, resuspended in BECs media consisting of DMEM-F12 supplemented with 10% Bovine platelet-poor plasma-derived serum (First Link), 10μg/ml Insulin-transferrin-sodium selenite, 10μg/mL gentamicin (Lonza), and 1ng/mL freshly added basic fibroblast growth factor (bFGF)

(PeproTech), and seeded in collagen type IV (0.15mg/ml)/fibronectin (0.05mg/ml)-coated Petri dishes or culture flasks. The BECs medium was supplemented with 4μg/mL puromycin for the first 3–4 days of culture to obtain pure cultures of BECs [44]. The BECs medium for culturing pBECs was additionally supplemented with chloramphenicol for the first three days of culture, due to the high occurrence of methicillin-resistant staphylococcus aureus (MRSA) (CC398) in Danish domestic pigs. BECs were maintained at 37˚C in 5% $CO_2$ in a humidified atmosphere. The growth of the BECs was analyzed in a Primo Vert phase-contrast microscope (Carl-Zeiss, Germany) equipped with an AxioCam ERc5s camera, and analyzed for brightness contract in ImageJ [45].

### Isolation of astrocytes

Astrocytes were isolated from mice, rats, and porcine brains according to previously published protocols [22, 28, 31]. Mouse and rat astrocytes were isolated from one to two, two-day-old C57BL/6 mice and Sprague-Dawley rat brains, respectively. The porcine astrocytes were isolated from the brains of six-month-old domestic pigs obtained from the local abattoir.

The mice and rat pups were decapitated and their brains were collected, while 1–2 grams of porcine brain tissue were collected. The brains/brain tissue was suspended in astrocyte media consisting of DMEM (Life Technologies), supplemented with 10% fetal calf serum (FCS) and 10μg/mL gentamycin. The tissue was mechanically dissociated with a Hypodermic needle 21G x 3 $^1/_8$"(Sterica B. Braun Medical A/S), filtered through a 40μm Corning cell strainer. The filtered cell suspension was seeded onto Poly-L-Lysine (500μg/ml) coated culture flasks, and maintained at 37˚C and 5% $CO_2$ in a humidified atmosphere. The cells were cultured for two to three weeks, with medium changes every third day, after which they were reseeded in poly-L-lysine coated 12 well culture plates and cultured for another one to two weeks prior to co-culturing with BECs. The medium for porcine astrocytes was also supplemented with chloramphenicol for the first three days. The astrocyte cultures also contain few microglia cells [22, 28], however, throughout the manuscript, the culture will be referred to as astrocytes.

### *In vitro* BBB model construction

The BECs were cultured either as a mono-culture or in a non-contact co-culture with astrocytes, referred to as a "co-culture" from here on. BECs were seeded at a density of approximately 100,000 cells/cm$^2$ on collagen/fibronectin double-coated hanging culture inserts (1μm pore size, 1.12cm$^2$) (In Vitro A/S) in 12 well plates and left to adhere and proliferate for 24 hours to obtain a confluent monolayer. To construct the co-culture of BECs with astrocytes, the hanging culture inserts with BECs were transferred to a 12 well plate containing astrocytes. To induce BBB characteristics in both models, the BECs media was supplemented with 250μM 8-(4-Chlorophenylthio)adenosine 3′,5′-cyclic monophosphate sodium salt (CTP-cAMP), 17.5μM RO-1724 (RO), and 550nM hydrocortisone (HC) in the upper chamber and 550nM HC in the lower chamber. In the co-culture model, the media composition of the lower chamber was a combination of BECs media and astrocyte conditioned media (1:1). Astrocyte conditioned medium is medium, which has been in contact with the astrocyte culture for at least 24 hours.

### Transendothelial electrical resistance measurements

The barrier integrity of the *in vitro* BBB models was assessed through measurements of TEER using Millicell ERS-2 epithelial volt-ohm Meter and STX01 Chopstick Electrodes (Merck Millipore). The TEER value for each hanging culture insert was obtained from an average of three individual measurements subtracted from the TEER value of a double-coated cell-free

hanging culture insert and multiplied by the area of the hanging culture insert ($1.12cm^2$). TEER was measured once a day after seeding of the BECs on the hanging culture inserts.

## Apparent permeability

The Papp of radiolabeled [3H]-D-Mannitol (PerkinElmer) was measured to further assess the barrier integrity of the different *in vitro* BBB models. 1µCi [3H]-D-Mannitol (Specific activity 15.9 Ci/mol) was added to the medium inside the hanging culture insert (donor compartment), and samples of 100 µl medium were collected from the lower chamber (receiver compartment) at 0, 15, 30, 60, 90, and 120 min, and replaced with 100 µl fresh medium. The cells were incubated at 37˚C under mild agitation. The samples were mixed with Ultima Gold™ liquid scintillation cocktail (PerkinElmer) and the radioactivity was counted in a Hidex 300SL liquid scintillation counter. The total amount of mannitol transported across the BECs was plotted against time and the slope at steady-state was calculated. Subsequently, Papp (cm/s) was calculated by dividing the slope with the area of the culture insert ($1.12cm^2$) multiplied by the initial concentration. For each filter insert the calculated Papp data were plotted against TEER. Cells from at least two different batches were used, resulting in 12–16 filter inserts for each culture condition and each species.

## Gene expression analysis

RNA was isolated from both mono- and co-cultured BECs from each species to evaluate the gene expression levels of tight junction proteins (*Cld1*, *Cld3*, *Cld5*, *Ocln*, *Zo1*, and *Zo2*), receptors (*TfR*), and efflux transporters (*P-gp* and *Bcrp*). RNA was isolated on the day of the expected highest TEER, thus RNA from mBECs and rBECs was isolated on day two or three, and RNA from pBECs was isolated on day three to four (Fig 1A). Each RNA sample consisted of a pool of cells from three hanging culture inserts and cells from at least two different batches were used, resulting in a total of six RNA samples. RNA was extracted using the GeneJET RNA Purification Kit (Life Technologies), according to the manufacture's guidelines. Genomic DNA was removed using DNase I enzyme (Life Technologies) before cDNA synthesis, which was carried out using the Maxima H Minus First-Strand Synthesis Kit (Life Technologies). 2.5ng cDNA and 10pmol of each primer (TAG Copenhagen) were added to the Maxima™ SYBR Green qPCR Master Mix in a final reaction volume of 20µL. For normalization purposes, β-actin (*Actb*) and hypoxanthine phosphoribosyltransferase 1 (*Hprt1*) were used as housekeeping genes, while non-reverse transcribed RNA and water served as negative controls. A list of primer sequences used in the qPCR can be found in Tables 1–3. qPCR was performed using the Stratagene Mx3000P™ QPCR system (Agilent Technologies) running the following program: 95˚C for 10 min, 40 cycles of 95˚C for 30 sec, 60˚C for 30 sec, and 72˚C for 30 sec. The relative expression of mRNA was calculated according to the Pfaffl-method with the mono-cultured BECs of each species serving as the calibrator sample (control) [46].

## Immunocytochemistry

Immunocytochemical stains were performed on mice, rats, and porcine astrocytes using the astrocyte marker glial-fibrillary acid protein (GFAP). Additionally, both mono- and co-cultured BECs were immunolabeled to evaluate the expression and localization of tight junction proteins (CLD5, OCLN, ZO-1, ZO-2), receptors (TfR), and efflux transporters (P-gp). The cells were washed twice in 0.1M phosphate-buffered saline (PBS), fixated for 5–10 min in 4% paraformaldehyde, and washed twice in 0.1M PBS. To block unspecific binding of the antibodies, the cells were incubated for 30 min in a blocking buffer consisting of 3% BSA and 0.3% Triton X-100 in 0.1M PBS, before the addition of primary antibodies (Table 4). All incubations

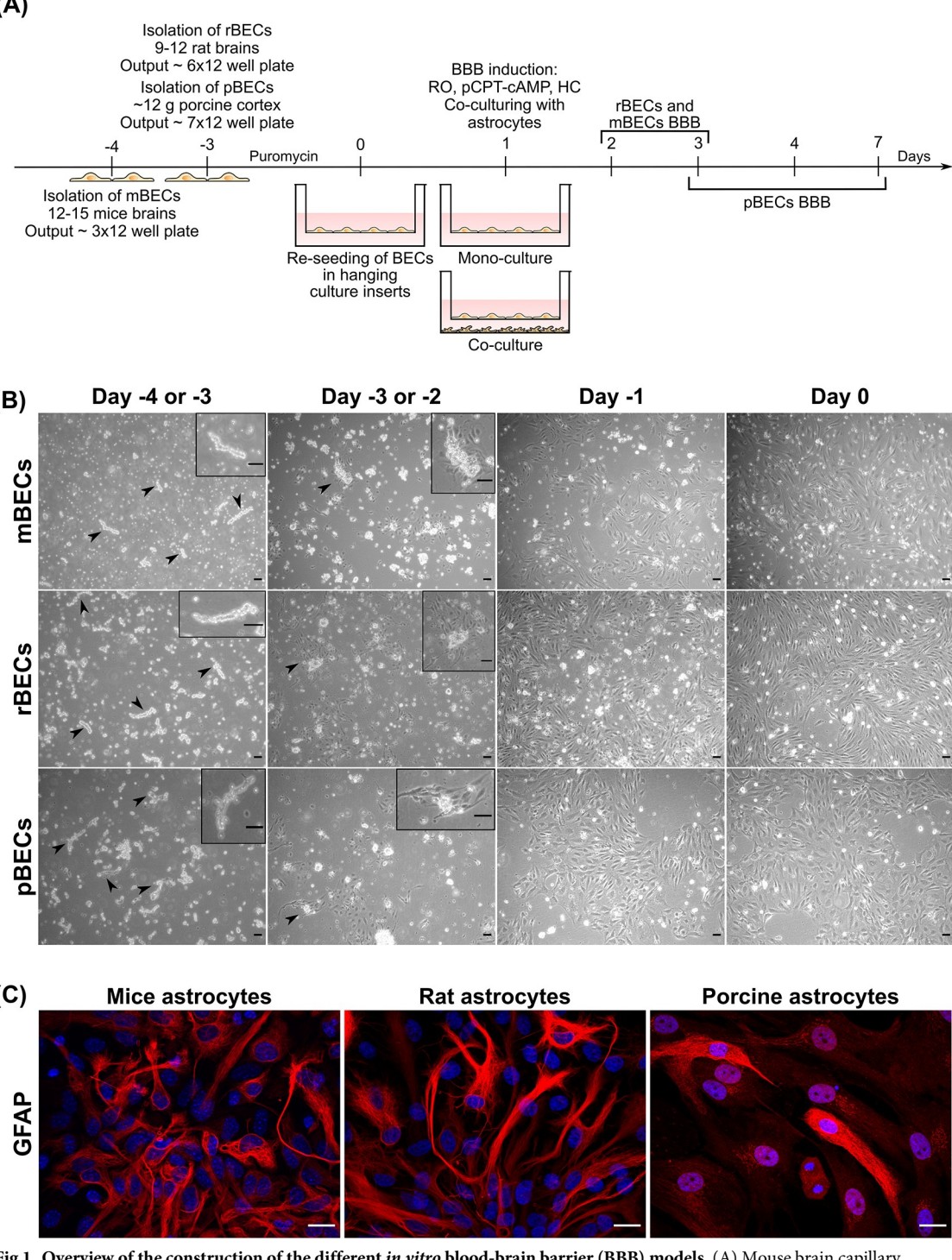

**Fig 1. Overview of the construction of the different *in vitro* blood-brain barrier (BBB) models.** (A) Mouse brain capillary endothelial cells (mBECs) are isolated on day -4, while both rat BECs (rBECs) and porcine BECs (pBECs) are isolated on day -3. The BECs are grown in a medium containing puromycin until day 0, to secure a pure BECs culture. On day 0, the cells are re-seeded on collagen IV/fibronectin-coated hanging culture inserts. 24 hours later, the BBB properties are induced by the addition of RO 20–1724 (RO), 8-(4-Chlorophenylthio)adenosine 3′,5′-cyclic monophosphate sodium salt (pCPT-cAMP), and hydrocortisone (HC), and the BECs are subsequently cultured as mono-culture or in co-culture with astrocytes isolated from mice, rat, or porcine brains. (B) Phase-contrast images of the isolated microvessels from day -4 or -3 to day 0 for mBECs, rBECs, and pBECs. The isolated microvessels (arrowheads and boxes) are recognized as small pearls on a string. After 24 hours the BECs start to proliferate from the microvessels, which is highlighted in the box and by the arrowheads. The cells reach a confluence level of 80–90% on day 0. Scale

bar = 20μM. (C) Immunofluorescent images of the primary astrocyte cultures isolated from mouse, rat, and porcine brains used for the creation of co-cultures. The astrocytes express the glial fibrillary acidic protein (GFAP; red), though at varying intensity, confirming the presence of astrocytes in these cultures. The nuclei are stained with DAPI (blue). Scale bar = 20μM.

were performed at room temperature under mild agitation. BECs were incubated with primary antibodies diluted 1:200–1:250 in blocking buffer for one hour and washed twice with washing buffer (blocking buffer diluted 1:50 in 0.1M PBS). Secondary goat anti-rabbit Alexa Fluor 488 (Invitrogen, Cat. No. A11034), donkey anti-mouse Alexa Fluor 488 (Invitrogen, Cat. No. A21202) or donkey anti-rat Alexa Fluor 488 (Invitrogen. Cat. No A21208) antibodies were diluted 1:200–1:250 in incubation buffer and added to the cells for one hour. Secondary antibodies were aspirated, and the cells washed twice in 0.1M PBS. The nuclei were stained with DAPI (1:500 in 0.1M PBS) and mounted on glass slides using fluorescent mounting media (DAKO) and examined in a fluorescence Observer Z1 microscope with ApoTome 2 under a Plan-Apochromat 40x/1.3 Oil DIC objective (Carl Zeiss). Captured images were processed using ImageJ [45].

## The functionality of the P-glycoprotein efflux transporter

To assess the functionality of the efflux transporter P-gp, the Papp of the radiolabeled P-gp substrate [3H]-digoxin (PerkinElmer) was measured in the luminal to abluminal (L-A) and abluminal to luminal (A-L) direction in the presence or absence of the P-gp inhibitor zosuquidar (ZSQ) (Selleckchem) for both mono- and co-cultured BECs from the three species. 0.4 μM ZSQ [47] or equal amounts of water were added to both the luminal (L) and abluminal (A) chamber 15 min before the addition of 1μCi/ml [3H]-digoxin to either the upper (L-A) or lower chamber (A-L). The cells were incubated at 37°C for two hours under mild agitation. Cell culture medium from the opposite chamber from where the [3H]-digoxin was added, was collected at 0, 15, 30, 60, 90, and 120 min, and replaced with equal amounts of fresh medium. Samples sizes of 50μl were obtained from the upper chamber, while 100μl samples were collected from the lower chamber. The samples were mixed with Ultima Gold™ liquid scintillation cocktail, and the radioactivity was counted in a Hidex 300SL liquid scintillation counter. The Papp for luminal to abluminal (L-A) and abluminal-luminal (A-L) was calculated as described for mannitol. The Papp was used to calculate the efflux ratio (ER), which refers to the ratio of Papp(A-L)/Papp(L-A). Cells from at least two different batches were used (n = 6).

**Table 1. Overview of the primers used for mBECs.**

| Mus Musculus | | | |
|---|---|---|---|
| **Gene** | **NCBI** | **FW** | **REV** |
| *Actb* | NM_007393.3 | CTGTCGAGTCGCGTCCACC | TCGTCATCCATGGCGAACTGG |
| *Hprt1* | NM_013556.2 | GTTGGATACAGGCCAGACTTTGTTG | GATTCAACTTGCGCTCATCTTAGGC |
| *Cld1* | NM 016674.4 | CACCGGGCAGATACAGTGCAA | ATGCACTTCATGCCAATGGTGGA |
| *Cld3* | NM_009902.4 | CCTCTATTCTGCGCCGCGAT | CGACTGCTGGTAGTGGTGACG |
| *Cld5* | NM_013805.4 | AGGATGGGTGGGCTTGATCCT | GTACTCTGCACCACGCACGA |
| *Ocln* | NM_008756.2 | GATTCCGGCCGCCAAGGTT | TGCCCAGGATAGCGCTGACT |
| *Zo1 (Tjp1)* | NM_001163574 | GAGACGCCCGAGGGTGTAGG | TGGGACAAAAGTCCGGGAAGC |
| *P-gp (Abcb1)* | NM_011076.2 | AGGTAGAGACACGTGAGGTCGT | AACATTGTAAGCACACTGACTGCTG |
| *Bcrp (Abcg2)* | NM_011920.3 | GGCCATAGCCACAGGCCAAA | GACAGCCAAGGCCCAATGGT |
| *TfR* | NM_011638.4 | CTATAAGCTTTGGGTGGGAGGCA | AGAATGCTGATCTGGCTTGATCCAT |

**Table 2. Overview of the primers used for rBECs.**

**Rattus Norveigus**

| Gene | NCBI | FW | REV |
|---|---|---|---|
| *Actb* | NM_031144.3 | CCTCTGAACCCTAAGGCCAACCGTGAA | AGTGGTACGACCAGAGGCATACAGGG |
| *Hprt1* | NM_012583,2 | TGCAGACTTTGCTTTCCTTGGTCA | TGGCCTGTATCCAACACTTCGAG |
| *Cld1* | NM_031699.2 | ATCGTGACTGCTCAGGCCATC | TACCATCAAGGCTCTGGTTGCC |
| *Cld3* | NM_031700.2 | ATTACCTGGCCTAGGAACTGTCCAA | TAGTTTGCCTGTCTCTGCCCACTAT |
| *Cld5* | NM_031701.2 | CTACAGGCTCTTGTGAGGACTTGAC | AGTAGGAACTGTTAGCGGCAGTTTG |
| *Ocln* | NM_031329.2 | CTGACTATGCGGAAAGAGTCGACAG | AGAGGAATCTCCTGGGCTACTTCAG |
| *Zo1 (Tjp1)* | NM_001106266.1 | GCCTGCCAAGCCAGTCCATT | ACTGTGAGGGCAACGGAGGA |
| *Zo2 (Tjp2)* | NM_053773.1 | CTGCGCGCTGACACTGCT | CTGTGCGCTGCAGAGTGCTT |
| *P-gp (Abcb1)* | NM_012623.2 | AATCAACAGTACACAGACCGTCAGC | CCAAAGTGAAACCTGGATGTAGGCA |
| *Bcrp (Abcg2)* | NM_181381.2 | GAGTTAGGCCTGGACAAAGTAGCAG | AATCAACAGTACACAGACCGTCAGC |
| *TfR* | NM_022712.1 | TGGATCAAGCCAGATCAGCATTCTC | TTTCTTCCTCATCTGCAGCCAGTTT |

**Table 3. Overview of the primers used for pBECs.**

**Sus Scrofa**

| Gene | NCBI | FW | REV |
|---|---|---|---|
| *Actb* | XM_003124280.2 | CAGAGCGCAAGTACTCCGTGTGGAT | GCAACTAACAGTCCGCCTAGAAGCA |
| *Hprt1* | NM_001032376.2 | AATGCAAACCTTGCTTTCCTTGGTC | GGCATAGCCTACCACAAACTTGTCT |
| *Cld1* | NM_001244539.1 | ATCCTGCTGGGACTAATAGCCATCT | CCATACCATGCTGTGGCAACTAAGA |
| *Cld5* | NM_001161636.1 | GTCTTGTCTCCAGCCATGGGTTC | GTCACGATGTTGTGGTCCAGGAAG |
| *Ocln* | NM_001163647.2 | GCCCATCCTGAAGATCAGGTGAC | CTCCACCATATATGTCGTTGCTGGG |
| *Zo1 (Tjp1)* | XM_021098896.1 | AAGCCTCCAGAGGGAGCATCTAA | ATATCTTCAGGTGGCTTCACTTGGG |
| *Zo2 (Tjp2)* | NM_001206404.1 | ACAGAGGTTGAACCCATCATCCAAC | AATTGTGTCCTTCAAGCTGCCAAAC |
| *P-gp (Abcb1)* | XM_003130205.2 | CGATGGATCTTGAAGAAGGCCGAAT | CCAGTTTGAATAGCGAAACATGGCA |
| *Bcrp (Abcg2)* | NM_214010.1 | GCTATCGAGTGAAAGTGAAGAGTGGCT | AACAACGAAGATTTGCCTCCACCTG |
| *TfR* | NM_214001.1 | TTGATGATGCTGCTTTCCCTTTCCT | CCATTCTGTTCAACTGAGGAACCCT |

**Table 4. Overview of primary antibodies used for immunolabeling.**

| Target: | Mouse astrocytes | Rat astrocytes | Porcine Astrocytes |
|---|---|---|---|
| GFAP | Rabbit anti-bovine-GFAP, polyclonal, Agilent (Cat. No. Z0334), AB_10013382 | | |
| | **mBECs** | **rBECs** | **pBECs** |
| CLD5 | Rabbit anti-human CLD5, polyclonal, Sigma Aldrich, (Cat.No. SAB4502981), AB_10753223 | | |
| OCLN | Rabbit anti-human OCLN, polyclonal, Millipore, (Cat. No. ABT146), | | |
| ZO-1 | Rabbit anti-human ZO-1, polyclonal, ThermoFischer Scientific (Cat.No. 61–7300) AB_138452 | | |
| ZO-2 | Rabbit anti-ZO-2, polyclonal, Invitrogen, (Cat No 71–1400), AB_2533976 | | |
| P-gp | Rabbit anti-human P-gp, polyclonal, Abcam (Cat. No. ab129450), AB_11157199 | | |
| TfR | Rat anti-mouse TfR (RI7217), monoclonal, produced in-house via the hybridoma technique Clone: R17217 (RI7 217.1.4) | Mouse anti-rat CD71 (OX26), monoclonal, Bio-Rad (Cat. No MCA155G) AB_2201358 | Rabbit anti-porcine TfR, polyclonal, MyBioSource (Cat. No. MBS2052130) |
| | | | Rabbit anti-human CD71, polyclonal, Proteintech (Cat. No 10084-2-AP) AB_2240403 |
| | | | Rabbit anti-human Transferrin Receptor, polyclonal, Abcam (Cat. No Ab84036) AB_10673794 |

### Cell size analysis

The size of the BECs from each species was analyzed using ImageJ. Two hanging culture inserts from two different batches (a total of four hanging culture inserts per culturing condition) were immunolabeled for a tight junction marker, as previously described, and examined in a fluorescence Observer Z1 microscope with ApoTome 2 under a Plan-Apochromat 40x/1.3 Oil DIC objective. Two images were acquired per hanging culture insert under blinded conditions, resulting in a total of four images per culture condition included in the analysis (a total of eight images per species). Captured images were corrected for brightness and contrast and all uncut cells on each image were marked using the free-hand tools, and the area measured using Image J. The average cell area per image, taken from all cells marked by the tight junction staining of each image, was calculated and used in the analysis, resulting in a sample size of four per culturing condition (n = 4).

### Statistical analysis

Results are shown as mean ± standard deviation (SD). All experiments were performed at least twice using cells from two different batches of BECs. No blinding was used, except when analyzing images to determine the size differences between mBECs, rBECs, and pBECs. Statistical analysis was performed on data from the gene expression analysis and in the analysis of cell size. All graphs and statistical analyses were made using GraphPad Prism (version 8) and a 0.05 significance level was used. Since none of the datasets used for statistical analysis, contained sample sizes above six, no test for normality was included. Instead, the RT-qPCR datasets were analyzed for equal variances between the groups of each species (mono- compared to co-culture) using an F-test. If the variances were insignificantly different, a parametric unpaired t-test was used; while datasets that had significantly different variances were analyzed using a non-parametric Mann-Whitney test. No statistical comparison was made across the species, except for data sets regarding cell size, which were analyzed using two-way ANOVA with Tukey's multiple comparisons post hoc test to test the difference between mono- and co-culturing within the same species and to analyze size difference between mBECs, rBECs, and pBECs.

## Results

### Isolation of microvessels, cell growth, and *in vitro* BBB establishment

BECs were successfully isolated from mouse, rat, and porcine brains and used to construct mono-cultures and co-cultures with both BECs and astrocytes isolated from the respective species (Fig 1A and 1C). The mBECs exhibited slower cell growth than the rBECs and pBECs and thus were cultured for an extra day before being re-seeded in the hanging culture inserts (Fig 1A). mBECs were therefore isolated on day -4, while rBECs and pBECs were isolated on day -3. The isolated microvessels are recognized as small pearls on a string, and after one day (day -3 for mBECs and -2 for rBECs and pBECs) in culture the BECs start to proliferate from the microvessels and grow into a monolayer reaching a confluence level of 80–90% on day 0 (Fig 1B). Astrocytes from each species were identified based on their expression of the glial fibrillary acidic protein (GFAP) (Fig 1C).

Despite many similarities between the BECs derived from the three species, some differences do exist, especially in the number of cells obtained from one single isolation. For one batch of mBECs, equal to 12 to -15 mice brains, enough cells for three 12 well plates with hanging culture inserts ($1.12cm^2$) are obtained. One batch of rBECs obtained from 9–12 rat brains results in enough cells for six 12 well plates with hanging culture inserts, while 12 grams of

porcine cortex is used for one batch of pBECs results in cells for seven 12 well plates with hanging culture inserts (Fig 1A). We have tried to increase the output from the mBECs isolation by increasing the number of mice brains to 18–20 brains; however, this did not result in a significantly higher output of mBECs.

## The BECs express tight junction proteins and form a tight barrier *in vitro*

*In vivo*, BECs are characterized by the expression of tight junction proteins restricting the paracellular transport of solutes from the blood into the brain. Thus, we wanted to investigate the gene expression of a subset of these tight junction proteins in the BECs cultured in mono-culture and in co-culture with astrocytes, to investigate if co-culturing affected the expression and whether differences in the gene expression pattern exits between the three species. When comparing the effects of co-culturing the BECs with astrocytes on the expression of various tight junction proteins, the expression pattern was somewhat similar across the three species (Fig 2A). The expression of *Cld1* decreased after co-culturing the BECs with astrocytes, and the same tendency was observed when analyzing the expression of *Zo2*. However, despite multiple attempts with different primers and tissues reported with high expression of *Zo2*, the expression of *Zo2* in mBECs remained non-detectable (Fig 2A). The same applied to the analysis of *Cld3* in pBECs, which were non-detectable. The expression of both *Cld3* and *Cld5* by mBECs and rBECs was unaffected by co-culturing, oppositely to the pBECs, which significantly increased the expression of *Cld5* after co-culturing (Fig 2A). Furthermore, the co-culturing of BECs caused a significant increase in the expression of *Zo1* (Fig 2A).

Next, we sought to confirm the protein expression of the major tight junction proteins CLD5, OCLN, ZO-1, and ZO-2 and visualize their location in the BECs using immunocytochemistry. The tight junction proteins are all expected to be located at the cell-cell junction between two adjacent cells. In all three species, both the mono-cultured and co-cultured BECs expressed CLD5, OCLN, and ZO-1 at the cell-cell junctions (Fig 2B). CLD5 staining was also seen in the cytosol of mBECs and rBECs, probably due to the transport of the protein in vesicles from the Golgi apparatus to the cell junctions. The expression of OCLN by mono-cultured rBECs was weakly expressed at the cell-cell junctions, compared to the co-cultured rBECs, which had a much more defined expression. In the mono-cultured rBECs and pBECs, OCLN was primarily located in the cytosol and perinuclear region, indicating a high synthesis of the protein, which was not seen in the co-cultures. This observation is consistent with the decrease seen at the gene level at least in the co-cultured rBECs. Astrocytes might, therefore, have an impact on the expression and location of OCLN. The gene expression of *Zo2* was non-detectable in the mBECs (Fig 2A), and mono-cultured mBECs displayed negligent labeling of ZO-2, however, co-culturing of the mBECs lead to a pronounced expression of ZO-2 at the cell-cell junction (Fig 2B). Oppositely, to the mBECs, both rBECs and pBECs cultured in mono- and co-culture displayed junctional expression of ZO-2 (Fig 2B). These results indicate that the influence of astrocytes on the expression of tight junction proteins by BECs to some extend is species-dependent. The protein levels of ZO-1 did not seem to be affected by the presence of astrocytes: however, the organization of ZO-1 seemed to be affected, resulting in a clearer and more organized localization at the cell-cell junction.

## Correlation between transendothelial electrical resistance and permeability

The expression of tight junction proteins by the BECs contributes to the BECs having high TEER and low permeability. Thus, the integrity of the *in vitro* BBB models was evaluated by measuring TEER and analyzing the Papp of mannitol, a small molecule (182 Da) that is not a ligand for any BECs transporters (Fig 3A and 3B). The addition of BBB induction factors on

**(A)**

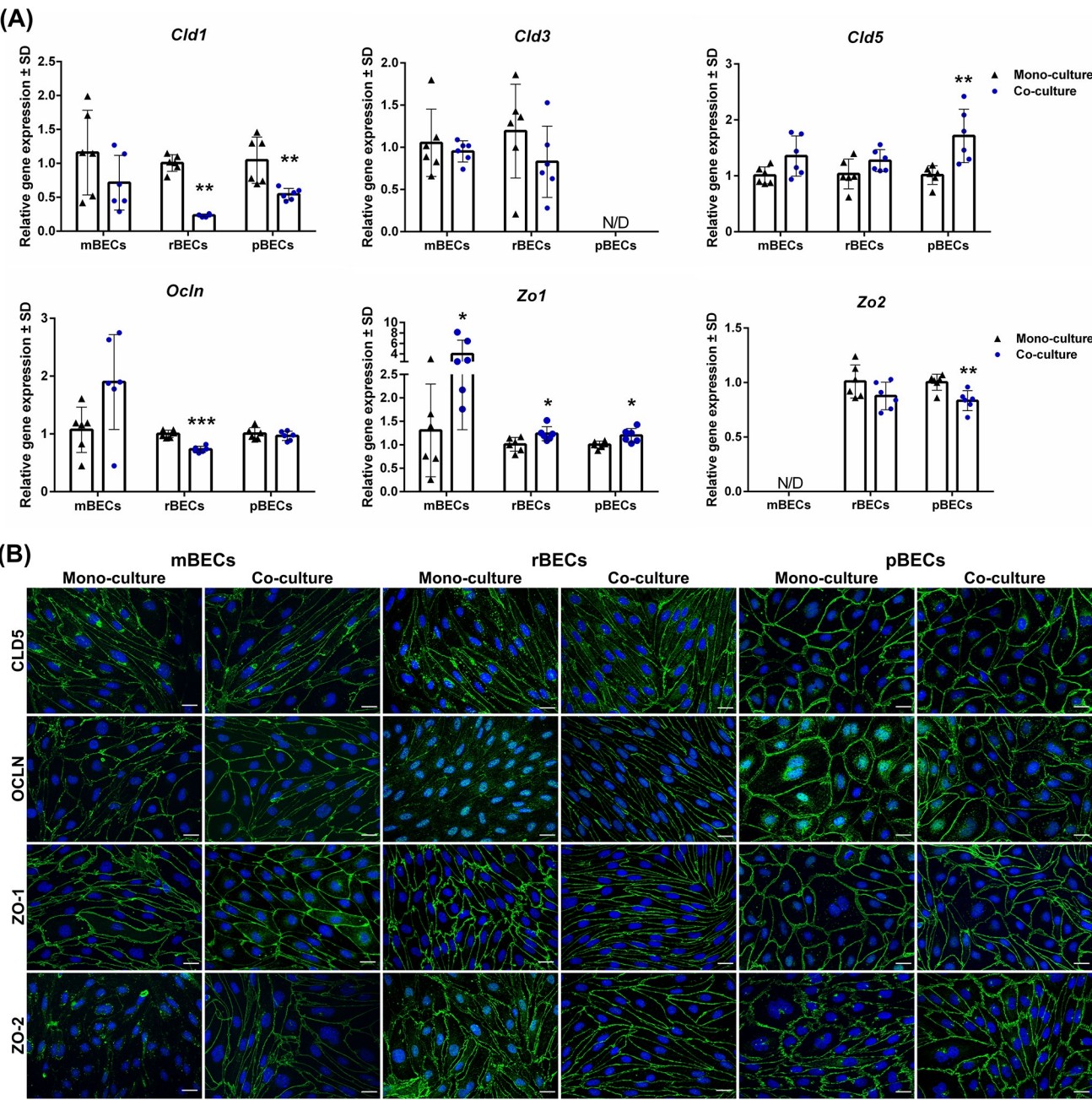

**(B)**

**Fig 2. Expression of tight junction proteins.** (A) Relative gene expression of claudin-1,-3,-5 (*Cld1*, *Cld3*, and *Cld5*), occludin (*Ocln*), and zonula occludens 1 and 2 (*Zo1*, *Zo2*) in mouse, rat, and porcine brain capillary endothelial cells (mBECs, rBECs, pBECs) cultured as mono-culture (black triangle) and as co-culture with primary astrocytes (blue circle). The expression pattern of the different tight junction proteins is relatively similar across the three species. Co-culturing the BECs with astrocytes decreases the expression of *Cld1 and Zo2* compared to mono-cultured BECs. Oppositely, a significant increase in the expression of *Zo1* is seen after co-culturing the BECs. *Cld3*, *Cld5*, and *Ocln* are unaffected by the culturing conditions except when rBECs and pBECs are co-cultured with astrocytes, where a significantly lower expression of *Ocln* or a significant increase in *Cld5* expression is seen, respectively. Despite multiple attempts, the expression of *Cld3* and *Zo2* is non-detectable in the pBECs and mBECs cultures, respectively. A change in gene expression between mono- and co-cultures for each species is analyzed using an unpaired t-test or non-parametric Mann-Whitney test, depending on the variance of the data. Data are shown as mean ± standard deviation (SD) (n = 6), $*p < 0.05$, $**p < 0.01$, $***p < 0.001$. (B) Immunofluorescent images showing green labeling at the cell-cell borders of CLD5, OLCN, ZO-1, and ZO-2 in mBECs, rBECs, and pBECs cultured in mono- or co-culture with astrocytes. The BECs express CLD5, OCLN, and ZO1 at the cell-cell interface independently of mono- or co-culturing. mBECs grown in mono-culture do not express ZO-2 at the cell borders, however, a clear expression along the cell border is seen in co-cultured mBECs. Both rBECs and pBECs have a clear expression of ZO-2 along cell borders independent of the culturing conditions. The nuclei are stained with DAPI (blue). Scale bar = 20μM.

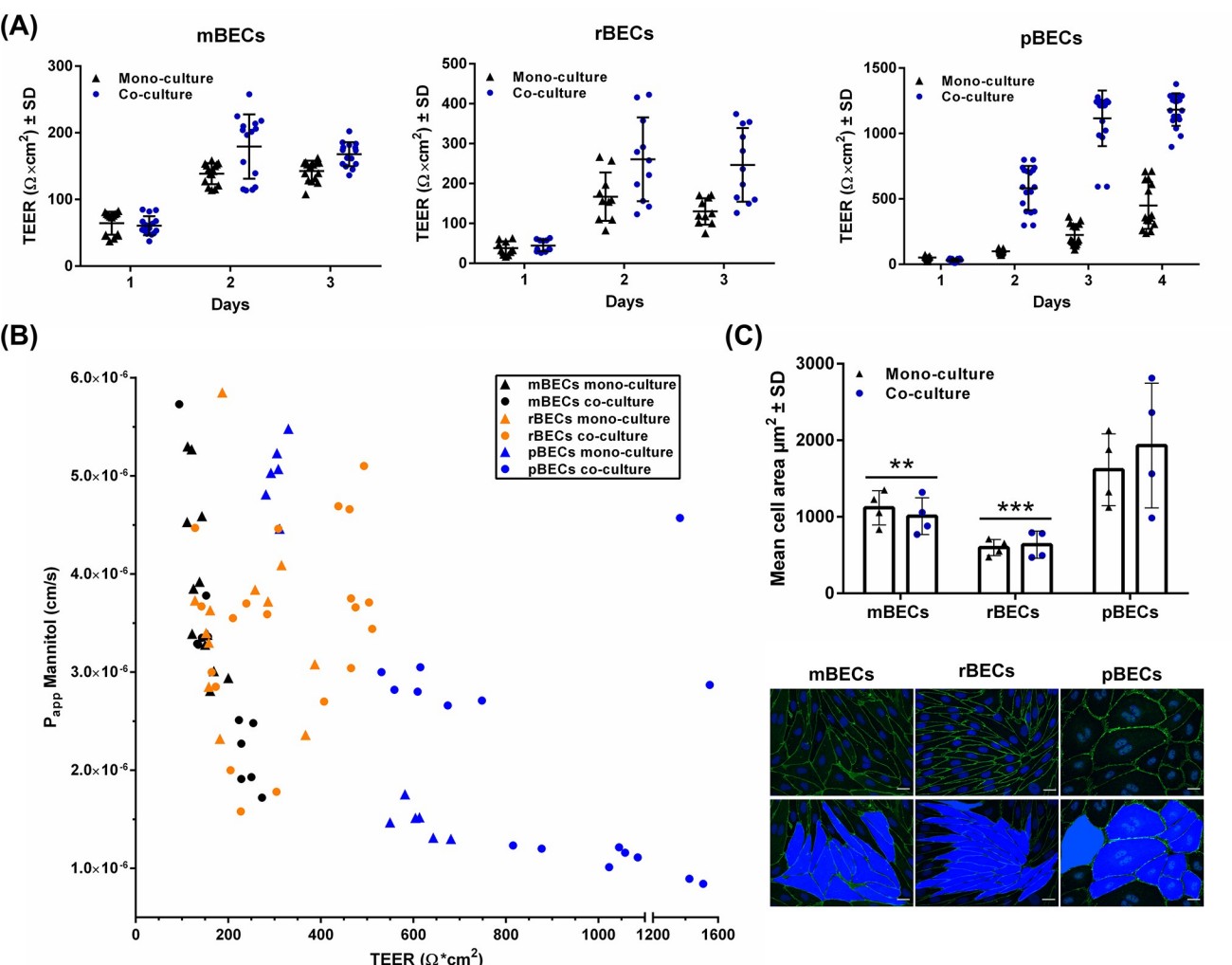

**Fig 3. Evaluation of the blood-brain barrier integrity.** (A) The transendothelial electrical resistance (TEER) ($\Omega^*cm^2$) of mouse, rat, and porcine brain capillary endothelial cells (mBECs, rBECs, pBECs) cultured in mono-culture (black triangles) and co-culture with astrocytes (blue circles). Co-culturing the BECs with astrocytes causes an increase in TEER compared to mono-cultured BECs. Each triangle or circle represents an individual culture insert showing a great variation in TEER across the inserts with identical culture conditions. Both mBECs and rBECs show a rapid increase in TEER after induction (day 1), peaking around day two to three, oppositely to pBECs, which have a slower increase in TEER peaking around day four. The data are illustrated as mean ± standard deviation (SD) (mBECs n = 15, rBECs n = 10–11, and pBECs n = 15–18). (B) The functional barrier integrity of the BECs was additionally evaluated by calculating the apparent permeability (Papp) of [3H]-D-Mannitol plotted against TEER across mBECs (black), rBECs (orange), and pBECs (blue) cultured in mono-culture (triangles) or co-culture (circles). The higher the TEER, the lower the Papp; however, in the TEER range of 150–650 $\Omega^*cm^2$ only small differences in the Papp are observed. mBECs n = 12, rBECs n = 12, and pBECs n = 12–16 for each culturing condition. (C) Quantification of cell area of mBECs, rBECs, and pBECs cultured in mono- and in co-culture. pBECs are significantly larger than both mBECs (p = **) and rBECs (p = ***), and co-culturing the BECs does not affect the cell size. The bottom panel shows representative images of how the area of the cells was quantified. Scalebar = 20μM. To analyze the difference in size between mono- and co-cultured BECs within the same species and to compare differences in size between species, a two-way ANOVA with Tukey's multiple comparisons post hoc test was used. ** *p< 0.01,* *** *p< 0.001.* Data are depicted as mean cell area μm² ± SD (n = 4).

day one caused a dramatic increase in the TEER (Fig 3A, day two), and co-culturing of the BECs resulted in a further increase in TEER (Fig 3A). However, a large variation in TEER exists from batch to batch but also when comparing the mBECs, rBECs, and pBECs. mBECs reached maximum TEER on day three, where mean TEER values for mono-cultured mBECs were 139.7 ± 15 $\Omega^*cm^2$ and co-cultured mBECs 183.2 ± 46.3 $\Omega^*cm^2$ (Fig 3A). rBECs reached

maximum TEER on day two with mean TEER values of $167 \pm 60.7 \; \Omega^*\text{cm}^2$ in mono-cultured rBECs and $260.6 \pm 105 \; \Omega^*\text{cm}^2$ for co-cultured rBECs. pBECs, on the other hand, had a slower increase in TEER reaching its maximum on day four (Fig 3A). pBECs cultured in mono-culture displayed mean TEER values of $449.1 \pm 178.4 \; \Omega^*\text{cm}^2$ and co-cultured pBECs reached mean TEER values of $1182.3 \pm 124.4 \; \Omega^*\text{cm}^2$.

As mentioned, the TEER values vary from batch to batch of isolated cells, thus all species can reach both lower or higher TEER values than that displayed in Fig 3A, which visualizes TEER values measured from individual filters of BECs from two different batches of isolated cells. mBECs and rBECs normally display the highest TEER values for two days, during which experiments are normally performed. After day three, the barrier integrity of mBECs and rBECs decreases. pBECs, on the other hand, maintain a stable TEER for a longer period. In this study, experiments on mBECs and rBECs were performed either on day two or three, while experiments on pBECs were performed on day three or four. The pBECs reached significantly higher TEER values than both of the rodent models, when comparing the different species on the day of highest TEER within the same culture condition (P<0.001 for mono-cultures, and co-cultures). No significant differences were found between the two rodent models (data not shown).

Besides TEER, the paracellular permeability was analyzed by measuring the Papp of [3H]-mannitol (Fig 3B). The Papp of mannitol was measured in the mono- and co-cultured mBECs, rBECs, and pBECs and plotted against TEER for each individual hanging culture insert. Since the different species of BECs display different TEER ranges, an ongoing discussion regarding choosing the most suitable *in vitro* BBB model for translational purposes is present both in the literature and in the general BBB community, with some arguments that the rodent models, displaying the lowest TEER values are leakier than the porcine and bovine models displaying significantly higher TEER values. As already stated, we show that TEER values were significantly higher in the *in vitro* BBB models established using pBECs compared to those established with mBECs and rBECs. We, therefore, set out to investigate whether this meant that mBECs and rBECs are more permeable than pBECs as indicated with lower TEER values. The Papp data were therefore plotted in the same graph for comparison (Fig 3B). TEER values for mBECs were in the range of 94 to $254 \; \Omega^*\text{cm}^2$, rBECs were in the range of 187 to $504 \; \Omega^*\text{cm}^2$, and the TEER value range for pBECs was on the other hand much wider, ranging from 281 to $1509 \; \Omega^*\text{cm}^2$. In the lower TEER ranges (from $150–650 \; \Omega^*\text{cm}^2$) there is no clear correlation between the permeability of [3H]-D-Mannitol and TEER. However, above $650 \; \Omega^*\text{cm}^2$ the permeability generally remains in the lower range. Looking at the data obtained from mBECs and to some extend also the pBECs, TEER inversely correlated with the Papp of [3H]-D-Mannitol. This pattern is, however, not seen in rBECs. All data points are within the range of $1.0^*10^{-6}$ to $6.0^*10^{-6}$ to (cm/s), which is within the normal range of that reported for *in vitro* BBB models based on primary cells [7, 16]. Therefore, even though large differences are seen in TEER, interestingly enough the permeability of [3H]-D-Mannitol is not highly variable between the different species. mBECs, which show the lowest TEER range, still display Papp in the same range as rBECs and pBECs, which both have higher TEER values. The Papp for pBECs in the highest range ($800–1500 \; \Omega^*\text{cm}^2$) is, however, generally lower than both mBECs and rBECs.

When looking at the immunocytochemical stainings of the mBECs, rBECs, and pBECs (Fig 2B), it is clear that the size of the BECs varies. We, therefore, hypothesize that the difference seen in TEER values among the three species might be somewhat explained by different sizes of the cells, with the argument that larger cells have fewer cell-cell junctions per filter insert and therefore a smaller area for which flux of ions can occur, contributing to higher TEER values. The area of the cells was, therefore, quantified by marking the cell-cell junctions by immunolabeling the tight junctions of BECs cultured in mono- and co-culture with

astrocytes (Fig 3C). The quantification of the cell area revealed that mBECs
($1,063.3 \pm 222.7$ μm$^2$) were larger and had a larger variation in the size compared to the rBECs
($617.9 \pm 135.8$ μm$^2$), while pBECs ($1,773.4 \pm 638.3$ μm$^2$) were significantly larger than both
mBECs and rBECs. Additionally, the pBECs displayed even larger variation in size with some
pBECs being roughly the same size as the mBECs and rBECs, while other pBECs were almost
three times as big. A phenomenon that was often seen in the pBEC cultures was multinuclear
cells, indicative of cell division. This was also sometimes seen in the mBEC cultures but rarely
in the rBEC cultures. No differences in size were observed between mono- and co-cultured
BECs within the same species. Representative images used for quantification of cell size are
shown in Fig 3C.

## The BECs display functional efflux transporters independent of mono- or co-culturing

*In vivo*, another important characteristic of the BECs is the presence of efflux transporters, as
these play an important role in protecting the brain from blood-borne lipophilic substances
that can passively diffuse across the cell membrane. Several different efflux transporters are
known, but the most studied are P-gp and BCRP. Therefore, we analyzed the gene expression
of *Pgp* and *Bcrp* in both mono- and co-cultured mBECs, rBECs, and pBECs (Fig 4A and 4B).
*Pgp* expression is significantly higher in mBECs upon co-culturing with astrocytes, while the
expression decreases significantly in rBECs when co-cultured. In pBECs, the expression is
unaffected by the culturing condition. The expression pattern of *Bcrp* in the BECs is on the
other hand highly similar among the different species showing significantly increased expression of *Bcrp* upon co-culturing (Fig 4C). The expression of P-gp was further confirmed on a
protein level by immunocytochemical staining (Fig 4C). P-pg was located in the cell cytoplasm
in all three species of BECs, but to some extend also at the cell-cell junction of pBECs, corresponding well to previous reports [25]. Despite the differences observed in the gene expression
pattern of P-gp in the BECs (Fig 4A), no major differences were observed on a protein level
(Fig 4C).

To investigate the functionality and possible species differences of the P-gp efflux transporter in the *in vitro* BBB models, the Papp in L-A and A-L of [3H]-digoxin, a P-gp substrate,
was measured with and without the influence of the P-gp inhibitor ZSQ. Both mono- and co-cultured BECs had an active P-gp efflux transporter, as the difference between Papp L-A and
A-L diminished in the presence of ZSQ and no difference was observed between the different
species (Fig 4D–4F). The presence of an active efflux transporter in the BECs was further supported by calculating the ER based on the Papp data (Fig 4D–4F). The different BECs all had
ER >two and the ER decreased when adding ZQS inhibiting P-gp, confirming a polarized
active efflux transport of lipophilic molecules from the brain to the blood (Fig 4D) [48]. The
functionality of P-gp was unaffected by mono- or co-culturing of the BECs of different species
(Fig 4G).

## Expression of TfR by BECs

Receptor-mediated transport across the BBB is one of the most studied mechanisms for active
transporting drugs across the BBB, and namely, the TfR has been extensively studied for
decades [43]. The expression of TfR was, therefore, investigated in mono- and mBECs, rBECs,
and pBECs co-cultured with astrocytes. The gene expression level of *Tfr* was unaffected by the
culturing condition with an identical expression pattern observed in all species (Fig 5A). The
expression of TfR was confirmed on the protein level using species-specific antibodies against
the mouse and the rat TfR. Unfortunately; we were unable to identify an antibody against the

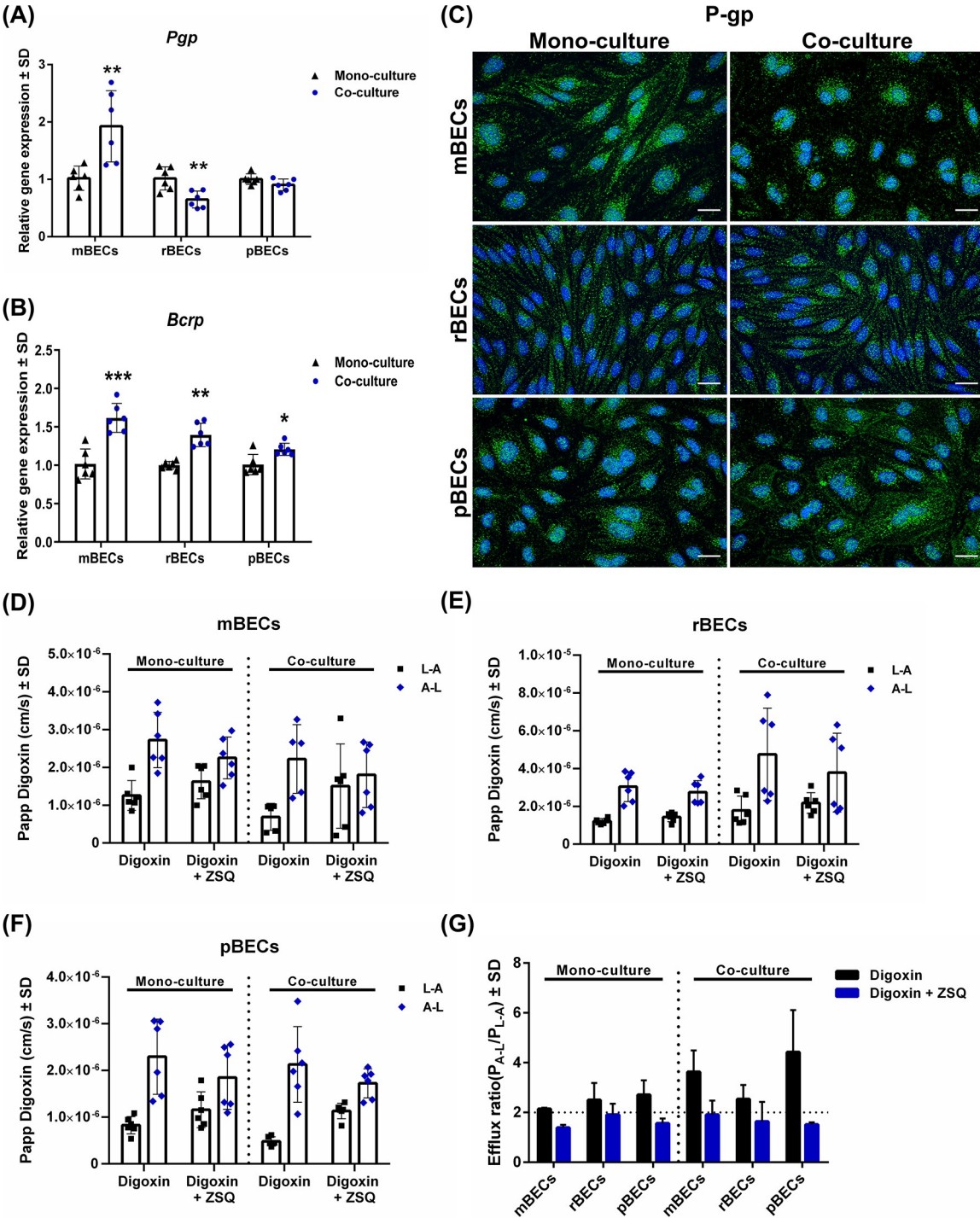

**Fig 4. Expression and functionality of efflux transporters.** (A+B) Relative gene expression of the efflux transporters P-glycoprotein (*Pgp*) and breast cancer resistant protein (*Bcrp*) in mouse, rat, and porcine brain capillary endothelial cells (mBECs, rBECs, pBECs) cultured in mono-culture (black triangle) and co-culture with astrocytes (blue circle). (A) The expression of *Pgp* increases significantly in co-cultured mBECs, decreases significantly in co-cultured rBECs, but remains unchanged in pBECs. (B) The expression of *Bcrp* is highly consistent in the BECs showing a significant increase in the expression upon co-culturing. (A+B) Changes in gene expression between mono- and co-cultures for each species were analyzed using an unpaired t-test or non-parametric Mann-Whitney test, depending on the variance of the data. Data are shown as mean ± standard deviation (SD) (n = 6), *$p < 0.05$, **$p < 0.01$, ***$p < 0.001$. (C) Immunofluorescent images showing green labeling of P-gp in the cell cytoplasm of BECs and also at the cell-cell borders in pBECs. Staining patterns are unaffected by the culturing condition. The nuclei are stained with DAPI (blue). Scale bar = 20μM. (D-F) The

functionality of the efflux transporter P-gp was evaluated by measuring the apparent permeability (Papp) to [3H]-digoxin, a P-gp substrate, with and without the P-gp inhibitor zosuquidar (ZSQ) in the luminal to abluminal (L-A) and abluminal to luminal (A-L) direction. The difference between L-A and A-L transport, which is caused by an active P-gp efflux transporter, diminishes when P-gp is inhibited. This applies to all three species independent of the culturing condition. Data are shown as mean ± SD (n = 6). (G) The efflux ratio (ER) Papp(A-L)/Papp(L-A) calculated from Papp (D-F) shows ER above 2 for all species and culture conditions, which decreases below 2 when P-gp is inhibited, indicative of an active polarized efflux transport function in the BECs. Data are shown as mean ± SD (n = 2).

porcine TfR that evoked reliable immunoreactivity. The TfR was primarily located in the cell cytoplasm, which corresponds well with the intracellular trafficking of the TfR from the granular endoplasmic reticulum to the cellular surface and endosomal-lysosomal compartment [38, 39].

## Discussion

In this study, we have constructed species-specific *in vitro* BBB models using primary BECs isolated from mouse, rat, or porcine brains, and compared these in relation to several major BEC characteristics, using the same protocol to isolate BECs from the three different species, making the cells and models highly comparable. However, we observed a prominent difference in the yield of BECs obtained per isolation, with the isolation of rBECs and pBECs giving a much higher cellular output than the mBECs. The low yield of mBECs has also been reported by other research groups [26, 49], suggesting that the feasibility and capacity of animal facilities, should be taken into consideration when choosing an *in vitro* BBB model using mBECs. Oppositely, the higher yield of rBECs and pBECs make these models cheaper and more

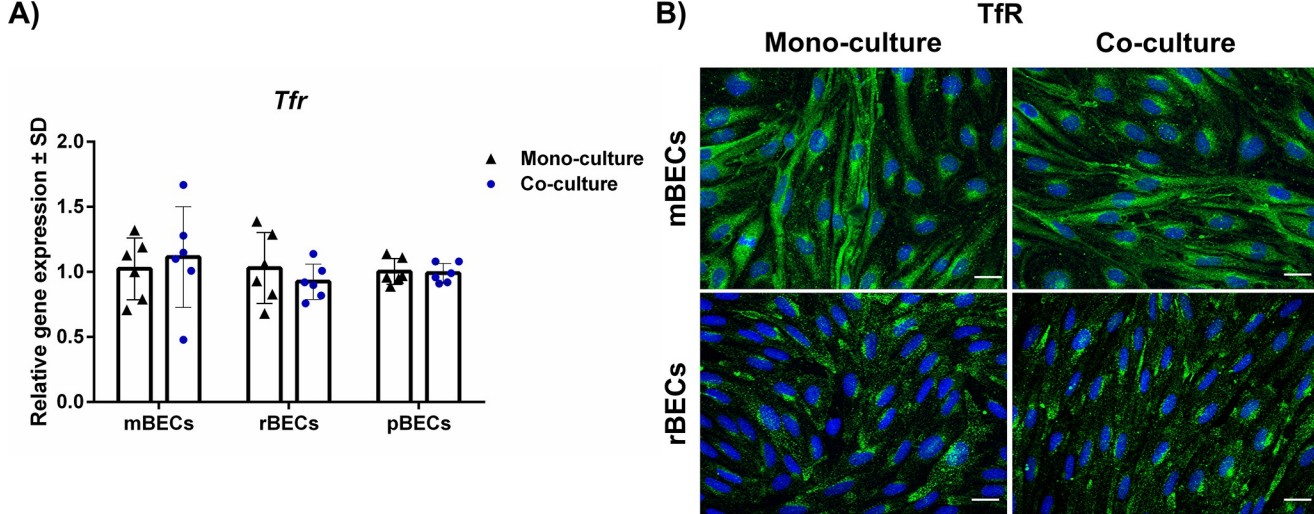

**Fig 5. Expression of the transferrin receptor (TfR).** (A) Relative gene expression of *Tfr*, which is a highly targeted receptor at the BBB for drug delivery purposes, in mouse, rat, and porcine brain capillary endothelial cells (mBECs, rBECs, pBECs) in mono-culture (black triangle) and co-culture with astrocytes (blue circle). *Tfr* is expressed in all three species independent of the culturing conditions. Changes in gene expression between mono- and co-cultures for each species were analyzed using an unpaired t-test or non-parametric Mann-Whitney test, depending on the variance of the data. No statistical difference was found. Data are shown as mean ± standard deviation (SD) (n = 6). (B) Immunofluorescent images showing green labeling of the TfR in the cell cytoplasm of both mBECs and rBECs, correspondingly well with the intracellular trafficking of the TfR. The nuclei are stained with DAPI (blue). Scale bar = 20μM.

convenient to work with, with pBECs being the cheapest to isolate as most abattoirs have porcine brains as a by-product and might donate the porcine brains for research.

## The expression pattern of various tight junction proteins

When comparing mono- and co-cultured BECs, it was evident that co-culturing the BECs with astrocytes influenced the gene expression of many tight junction proteins. However, the effect was not similar across mBECs, rBECs, and pBECs, suggesting differences in the complexity by which the tight junction proteins are organized. As an example, *Cld5* was upregulated, and *Zo2* downregulated in pBECs upon co-culturing, while these two proteins remained unaffected in rBECs. In mBECs, *Cld5* was likewise unaffected by co-culturing which is in good accordance with the literature, showing that the expression of *Cld5* is regulated by the addition of BBB induction factors like RO, CTP-cAMP, and HC [28, 50, 51]. *Zo2* was in our setup nondetectable at the gene level in mBECs, probably due to low expression, which was confirmed by ZO-2 immunolabeling showing negligent expression of ZO-2 in the mono-cultured mBECs. This contradicts the findings by Stamatovic et. al., showing immunolabeling of ZO-2 in mono-cultured mBECs [52]. We were, however, able to see ZO-2 immunolabeling after coculturing, suggesting that in our setup mBECs expression and organization of ZO-2 are dependent on the presence of astrocytes. We were unable to determine the *Cld3* gene expression in pBECs in our setup. However, CLD3 expression in pBECs has previously been confirmed by immunolabeling [33], suggesting that this protein is also expressed by pBECs.

The difference in gene expression levels between mono- and co-cultured BECs or between the species could not necessarily be translated to the protein level. The levels of protein did not seem to increase or decrease as suggested by the gene expression analysis, however, instead, the effect of co-culturing with astrocytes seemed to influence the organization and maturation of the tight junction proteins. This corresponds well with the fact that both the integrity and complexity of the tight junctions are increased after co-culturing with astrocytes [53, 54]. The influence of astrocytes on the tight junction expression was most prominent in mBECs and rBECs, and most obvious for the ZO-1 distribution at the cell-cell junctions. In mono-cultured mBECs and rBECs, the organization of ZO-1 was irregular and did not completely close the gap between the cell-cell junctions, while co-culturing the BECs resulted in the organization of ZO-1 becoming more uniform without gaps between the cells, suggesting that the presence of the astrocytes affected the maturation and organization of ZO-1 in the mBECs and rBECs. The same phenomenon is to some degree also observed in pBECs, but not as obvious. The effect of astrocytes on the expression of ZO-1 in BECs is probably through astrocytic secretion of sonic hedgehoc, as downregulation of sonic hedgehoc in astrocytes causes downregulation of ZO-1 in BECs and a decreased barrier integrity [14]. The addition of the BBB induction factor cAMP is also known to be able to increase the expression of tight junctions proteins [28, 54]. Furthermore, when comparing previously published immunolabelings of ZO-1 in rBECs cultured in monolayers in well plates, without the presence of neither astrocytes nor tight junctions inducing factors, the organization of ZO-1 seems to be highly affected by the addition of the tight junction inducing factors [55], suggesting that cAMP and HC is also important for the organization of this protein.

The distribution of OCLN was likewise affected by astrocytes, especially in the rBECs and pBECs. Immunolabeling of OCLN in the mono-cultures was seen in the cytoplasm and perinuclear areas indicating synthesis of OCLN, but when co-cultured, OCLN redistributed to the cell-cell junctions. On a gene level, *Ocln* was downregulated in the rBECs, which suggests that astrocytes affect the maturation and distribution of OCLN, by decreasing the synthesis and increasing the organization of OCLN at the cell-cell junctions. The regulation of OCLN in

BECs by astrocytes are also proposed to be through astrocytic secretion of sonic hedgehoc [14], and that the influence of astrocytes is through the secretion of soluble factors supported by several studies using astrocyte conditioned media, which is shown to be sufficient in inducing BBB characteristics in BECs [34, 56]. Together these studies also support the use of non-contact co-culture models, with astrocytes cultured in the bottom chamber, as used in the present study.

## Construction and integrity of the *in vitro* BBB models

The two types of *in vitro* BBB models used in this study were a simple mono-culture and the non-contact co-culture model. Therefore, only the effect of the astrocytes on the different species of BECs was investigated and compared, leaving the significance of pericytes for future studies. The triple culture model has been referred to as the most *in vivo* like model, as this model greatly mimics the anatomical structure of the BBB [29]. However, setting up the triple culture model is complex, which is probably why this model is not commonly used, despite it has been developed and characterized for all three species [22, 28, 29, 31, 57]. The mono-culture and the co-culture models are on the contrary easy to construct and therefore also the preferred models across different laboratories [32, 33, 37, 38, 40, 55, 58–61]. Future analysis of the significance of pericytes in *in vitro* BBB models should include analysis of the vesicular transport in the BECs as the presence of pericytes during embryogenesis decreases the vesicular transport and contribute to the formation of tight junctions [10, 62]. Analysis of vesicular transport has, therefore, not been included in the present study.

Oppositely to the small variations in the tight junction expression between mBECs, rBECs, and pBECs large variations in the TEER range were observed. The co-cultured mBECs reached TEER values around 150–250 $\Omega^*cm^2$ while co-cultured rBECs reached values around 150–400 $\Omega^*cm^2$ and co-cultured pBECs had TEER values between 500–1500 $\Omega^*cm^2$. This is in good accordance with the literature reporting similar TEER values for co-cultured mBECs, rBECs, and pBECs [29, 58–60]. However, to obtain a functional assessment of the tightness of the barriers, we wanted to further compare the TEER with the Papp of a small hydrophilic molecule across the different models. Since the pBECs have the highest TEER values, we hypothesized that pBECs would be less permeable, indicated by a lower Papp. We chose to investigate the Papp of the small molecule mannitol (180 Da), as this is a widely used marker for evaluating the functional tightness [7, 16, 25]. The Papp was in the range of $1$–$6^*10^{-6}$ and as TEER increased the Papp decreased in accordance with other studies [7, 16, 25]. However, in the TEER range of 150–650 $\Omega^*cm^2$, there was no clear correlation between permeability and TEER, suggesting that mBEC cultures with a TEER of 200 $\Omega^*cm^2$ are equally as impermeable as pBEC culture with a TEER value of 650 $\Omega^*cm^2$. Therefore, it cannot be postulated that BECs from species reaching high TEER values are significantly more impermeable for small hydrophilic molecules than BECs from species reaching lower TEER values at least when TEER is below 650 $\Omega^*cm^2$. This underlines the importance of TEER measurements being supported by permeability measurements. We propose that some of the differences in TEER and Papp observed between mBECs, rBECs, and pBECs are caused by cell size differences, with the pBECs being significantly larger than the mBECs and rBECs. Since pBECs are significantly larger, the number of cells per hanging culture insert must be fewer, and thereby the cell-cell junctions, through which molecules can pass, will be equally less. Additionally, since pigs belong to the higher species compared to rodents, they might have a more complex organization of tight junction proteins causing them to reach higher TEER values.

## Can *in vitro* BBB models be used for translational purposes?

An important characteristic of BECs *in vivo* is the expression of efflux transporters. We show that our *in vitro* BBB models have polarized functional P-gp efflux transporters and that the function is independent of mono- and co-culturing. The presence of functional P-gp efflux transporters in *in vitro* models using primary BECs is also reported by other research groups [29, 59, 63] and extensively reviewed in Helms et al; 2016 [7], supporting that our *in vitro* models are suitable for screening of small molecular drug transport and as all three species had a polarized active P-gp transport, the choice of model can be independent of the species.

The TfR is the most studied target on the BBB for the delivery of bispecific antibodies and nanoparticles [43]. We found equal gene expression of *Tfr* by mono- and co-cultured mBECs, rBECs, and pBECs. However, positive immunolabeling of TfR was only reported for mBECs and rBECs cultures. Monoclonal antibodies specific for the rodent TfR are known and widely used both *in vitro* and *in vivo*. The OX26 antibody is highly specific against the rat TfR while several specific monoclonal antibodies, e.g. the RI7217 antibody, have been developed to target the mouse TfR [39, 41, 43, 64]. However, to our knowledge, no such monoclonal antibody targeting the porcine TfR has been developed and used in drug delivery studies targeting the TfR similar to those published for rats and mice. Only a few antibodies specifically developed against the porcine TfR are commercially available, but since the porcine TfR might be somewhat similar to the human TfR, we also tested antibodies specific for the human TfR receptor on the pBEC cultures. We were, however, unable to identify a satisfying TfR antibody that labeled the porcine TfR receptor; for an overview of the tested antibodies, see Table 4. *In vitro*, drug delivery studies targeting the TfR are mainly performed using the rodent *in vitro* BBB models [39–41, 43]. The use of the *in vitro* BBB models based on mBECs and rBECs are preferred due to the high translatability to subsequent *in* vivo studies using both healthy and disease models.

## Animal vs human *in vitro* BBB models

In the present study, the focus is on BBB models constructed with primary mouse, rat, and porcine cells, since brain tissue from various animal sources is easy to obtain, and the cells maintain many of the important BBB characteristics. However, human primary cells constitute a theoretic optimal translational basis for a BBB model eliminating transgene hurdles, yet the access to these cells is limited. The few models using primary human brain endothelial cells (hBECs) are often obtained from biopsies from epilepsy-patients, thus impeding the foundation of a normal BBB [65–67], or from commercial vendors, which only have limited information on the source of the cells [7].

Primary hBECs generally express important BBB characteristics, like tight junction proteins (OCLN, CLD5, and ZO-1), and functional efflux transporters [7, 65]. The yield of primary hBECs has been reported to be $1^*10^6$ cells isolated from 5-10mm$^3$ fresh brain tissue [65]. Others have isolated from 4 to $7^*10^6$ cells from 1–13 grams of surgically removed fresh brain tissue, and $2^*10^6$ to $2^*10^7$ cells from autopsy material [66]. We obtained approximately $4^*10^6$, $8^*10^6$, $9.4^*10^6$ cells per isolation respectively for mice, rat, and porcine brains, as stated earlier. Therefore, the yield obtained from our animal models is higher compared to that obtained from fresh human tissue, and without limited access. In many of the studies constructing *in vitro* BBB models from primary hBECs, none or only the relative TEER values are provided [65, 67]. However, a single study using mono-cultured primary hBECs reported TEER values around 1000–1200 $\Omega^*$cm$^2$ [67]. Looking at the immunocytochemical stains of the hBECs they vary in size, with some being very large and others much smaller [65, 67], resembling what we observed in the pBECs cultures. We found that the cell size might influence the TEER range,

with larger cells, like pBECs, displaying higher TEER values, however, this is not necessarily equal to a much lower permeability as shown for our pBECs models. Not surprisingly, based on TEER and cell size primary hBECs are more similar to the pBECs than the mice and rBECs, but as we have shown earlier in this study, this is not necessarily equal to the cells being superior to the rodent models with respect to the Papp of small molecules.

Moreover, in recent years stem cell technology has evolved enabling the creation of human BBB models based on human pluripotent stem cells and human cord blood-derived stem cells [68–71]. Both types of stem cell-based BBB models show promising BBB characteristics including functional efflux transporters [68–71]. The models based on human pluripotent stem cells show TEER values around 200–3000 $\Omega^*cm^2$ for mono-cultures, around 1100 $\Omega^*cm^2$ with astrocyte conditioned media, and around 1450–3000 $\Omega^*cm^2$ when co-cultured with rat astrocytes or human pericytes [68, 69, 71]. When the BBB phenotype is enhanced by the addition of retinoic acid, the TEER values can be increased even further to approximately 5000 $\Omega^*cm^2$ [72]. A Papp of 0.5–0.6$^*10^{-6}$ cm/s for sucrose has been reported for the human pluripotent stem cell models, which are in the lowest range of all reported values obtained using *in vitro* BBB models [7, 69, 71]. TEER values of only 70–160 $\Omega^*cm^2$ depending on culture conditions have been reported using hBECs derived from blood cord stem cells, with corresponding Papp values of 10–20$^*10^{-6}$ cm/s for lucifer yellow, which is in the higher range but corresponds well to the low TEER values [68–71]. Based on measures of integrity, the BBB models constructed from hBECs derived from blood cord stem cells still need to be refined, whereas the BBB models constructed from hBECs derived from pluripotent stem cells seem highly promising. However, compared to the use of primary cells of animal origin, these models are very time-consuming to establish, and the line-to-line and batch-to-batch variability in yield and cellular phenotypes complicate the robustness of the process of differentiation from the human pluripotent stem cells [68, 70].

A few immortalized human BEC cell lines have been established, but often they lack crucial BBB characteristics and are rarely fully characterized [73]. The immortalized human microvascular endothelial cell line hCMEC/D3, established by Weksler et al in 2005 [74], remains one of the most characterized and highly studied cell lines for the construction of a human *in vitro* BBB model [7]. The hCMEC/D3 cell line expresses many of the important tight junction proteins however, the expression of CLD5, OCLN, and JAM2 is low at both the gene and protein levels [73, 75]. Especially the expression of CLD5 is critical for maintaining a low permeability towards small molecules (<800 Da) [76], however, the low expression of the tight junction proteins in these cells can be increased by optimizing the culture conditions, e.g. by the addition of HC [74, 75, 77]. Depending on the culture conditions TEER values observed for hCMEC/D3 are normally in the range of 30–300 $\Omega^*cm^2$ [13, 35, 73, 77], and permeability studies using hCMEC/D3 reveal Papp for mannitol to be 24.7$^*10^{-6}$ cm/s [78]. This is 20 times higher than the Papp reported in the present study, making all of our three animal models superior to the hCMEC/D3 cell line concerning the Papp to mannitol. The low barrier integrity reported with the hCMEC/D3 cell line might be due to the low expression of CLD5 in these cells, as mentioned previously [77]. It is well known that immortalized cell lines rarely reach TEER values and low Papp values comparable to those obtained using primary cells, however, the hCMEC/D3 express functional and polarized efflux transporters, like P-gp and BCRP, comparable to the primary cells, making them useful for studying drug interactions at the BBB [73, 79].

## Choosing the right *in vitro* model for future BBB modeling

The perfect *in vitro* BBB model would probably in most minds be of human origin, express all the important BBB characteristics found *in vivo*, be low-cost, non-time consuming, easy to

culture, and highly reproducible. However, at the moment, none of the human *in vitro* BBB models can fulfill all of these criteria, why the use of animal *in vitro* BBB models is still justified, as they display many characteristics of what would be considered an optimal model.

Current models all have advantages and disadvantages, why it is important to consider these when planning future experiments. As stated previously, there has been an ongoing discussion regarding choosing the optimal in vitro BBB model solely based on TEER values, since these vary significantly between different species. Instead, as also reported here, the Papp should be taken into account as well since the differences in TEER might solely be due to size differences between cells from rodents and higher animal species like the porcine. Instead one should consider choosing a model that would add to the consistency between *in vitro* and *in vivo* analysis. The *in vitro* BBB models constructed from mice would be more favorable to use when planning to move forward with *in vivo* studies using e.g. transgenic mice models. When studying potential drug interaction with the efflux transporters before moving into human clinical trials, it makes perfect sense to choose among the human *in vitro* BBB models, despite these do not express the necessary characteristic for sufficient barrier integrity. Likewise, when targeting the TfR for drug delivery across the BBB, it might be favorable to use one of the rodent models, since well-characterized antibodies have been developed against the TfR in these models. Despite the high output of cells and the easy and cheap access to porcine brain tissue, some gene and protein analyses are still highly challenged. Therefore, as no model seems superior to the others, the choice of *in vitro* BBB model should primarily be based on the purpose of the study, rather than a wish to obtain the highest TEER values.

## Conclusion

Based on the results from the present study, we conclude that not many differences exist when comparing mBECs, rBECs, and pBECs. However, the yield and translatability from *in vitro* to *in vivo* studies are parameters to consider. pBECs would be easy and cheap to use for high throughput *in vitro* drug delivery studies due to low costs and high cell yield, as opposed to mBECs. Isolation of mBECs result is the lowest cellular yield, but many *in vivo* disease mouse models exist enabling translation to *in vivo* research. This is opposed to the pBECs, which results in a high yield of cells but the translation to *in vivo* transgenic pig models is much more complicated. The use of pBECs is also complicated due to e.g. the availability of specific antibodies targeting porcine proteins. Furthermore, we find that the significantly higher TEER measurements obtained with the porcine *in vitro* BBB model are not necessarily translatable to a much less permeable cell layer compared to *in vitro* BBB models of mice or rat origin, as the Papp of [3H]-D-Mannitol is low for all three models. The large variation in TEER could instead be due to a significantly larger cell size of the pBECs compared to the mBEC3s and rBECs or be explained by a more complex organization of tight junction proteins in pBECs causing them to reach significantly higher TEER values. Finally, co-culturing BECs with astrocytes is preferable to maintain as many *in vivo* characteristics as possible and highly affect the gene expression pattern of the tight junction proteins. As stated earlier, the choice of *in vitro* BBB model should be based on the aim of application rather than false assumptions of high TEER being equal to a superior model.

## Acknowledgments

The authors wish to thank Hanne Krone Nielsen and Merete Fredsgaard for their excellent technical assistance.

## Author Contributions

**Conceptualization:** Maj Schneider Thomsen, Nanna Humle, Torben Moos, Annette Burkhart, Louiza Bohn Thomsen.

**Formal analysis:** Maj Schneider Thomsen, Annette Burkhart.

**Funding acquisition:** Maj Schneider Thomsen, Nanna Humle, Torben Moos, Annette Burkhart, Louiza Bohn Thomsen.

**Investigation:** Maj Schneider Thomsen, Nanna Humle, Eva Hede, Annette Burkhart, Louiza Bohn Thomsen.

**Methodology:** Maj Schneider Thomsen, Nanna Humle, Annette Burkhart, Louiza Bohn Thomsen.

**Project administration:** Maj Schneider Thomsen, Annette Burkhart, Louiza Bohn Thomsen.

**Resources:** Maj Schneider Thomsen, Nanna Humle, Torben Moos, Annette Burkhart, Louiza Bohn Thomsen.

**Supervision:** Maj Schneider Thomsen, Torben Moos, Annette Burkhart, Louiza Bohn Thomsen.

**Validation:** Maj Schneider Thomsen, Nanna Humle, Eva Hede, Annette Burkhart, Louiza Bohn Thomsen.

**Visualization:** Maj Schneider Thomsen, Annette Burkhart.

**Writing – original draft:** Maj Schneider Thomsen, Nanna Humle, Annette Burkhart, Louiza Bohn Thomsen.

**Writing – review & editing:** Maj Schneider Thomsen, Nanna Humle, Eva Hede, Torben Moos, Annette Burkhart, Louiza Bohn Thomsen.

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
