## [Decision Letter · Decision Letter 0]

11 Aug 2020

PONE-D-20-21385

The blood-brain barrier studied in vitro across species

PLOS ONE

Dear Dr. Burkhart,

Thank you for submitting your manuscript to PLOS ONE. After careful consideration, we feel that it has merit but does not fully meet PLOS ONE’s publication criteria as it currently stands. Therefore, we invite you to submit a revised version of the manuscript that addresses the points raised during the review process.

As indicated in the Decision, major revision, mainly due that both reviewers agreed that the necessity to increase the impact of the manuscript can be achieved by a deeper examination of the model or the inclusion of a well characterized  human model to compare them.  Both I think are essential.

We look forward to receiving your revised manuscript.

Kind regards,

Eliseo A Eugenin, Ph.D.

Academic Editor

PLOS ONE

Additional Editor Comments:

Dear Dr. Burkhart

Thank you for submit your manuscript to PLOSone. As indicated in the Decision, major revision, mainly due that both reviewers agreed that the necessity to increase the impact of the manuscript can be achieved by a deeper examination of the model or the inclusion of a well characterized human model to compare them. Both I think are essential.

Best regards

Eliseo

Journal Requirements:

Reviewers' comments:

Reviewer's Responses to Questions

**Comments to the Author**

1. Is the manuscript technically sound, and do the data support the conclusions?

Reviewer #1: Yes

Reviewer #2: Partly

2. Has the statistical analysis been performed appropriately and rigorously? 

Reviewer #1: Yes

Reviewer #2: Yes

3. Have the authors made all data underlying the findings in their manuscript fully available?

Reviewer #1: Yes

Reviewer #2: Yes

4. Is the manuscript presented in an intelligible fashion and written in standard English?

Reviewer #1: Yes

Reviewer #2: Yes

5. Review Comments to the Author

Reviewer #1: Thomsen and coauthors have evaluated property of the in vitro Blood brain barrier (BBB) generated from the three different species, mouse, rat and porcine. Analyzing the tight junction protein expression, brain endothelial barrier permeability (Papp and TEER assay ) and expression of Pgp efflux transporter and transferrin receptor, they found significant differences in TEER values and brain endothelial barrier tightness between porcine comparing with rat and mouse BBB models, suggesting that all models have limitation and should be utilized based on the propose of study.

This is a very good study on an important topic. In general, the manuscript is well prepared, and the experiments are carefully designed and mapped out. Data is convincing and well-presented and discussed. The manuscript is very well written and statistical analysis is very rigorous. However, there are several concerns regarding this study which should be addressed:

1. The authors should take in consideration that expression pattern of the tight junction (Tj)protein is different during the culturing/co-culturing and forming the barrier in vitro. It is not clear from presented data when was obtained samples, when the cells are confluent, or barrier is formed (5-7 days of culturing). It will be important to delineate the expression pattern of Tj mRNA expression in all three BBB models.

2. Regarding the expression pattern of Tj proteins it will be important to show the dynamic of protein expression together with mRNA.

3. Fig 3C is very important but it is very difficult to follow. Suggestion will be to split figure by species, keeping the same value on both x and y axis for easier comparison.

4. The authors completely neglected the role of pericytes in the in vitro BBB coculture model. Pericytes were showed to play critical role for the barrier integrity/permeability. The current property could be only limited on astrocytes role. This should be addressed.

5. One of the major goals of the study should be to compare these models to human in vitro BBB model. This will gain more translator character for the obtained results.

Reviewer #2: The study compares barrier and enzymatic properties of monocultures of brain endothelial cells isolated from mouse, rat, and porcine brains as well as co-cultures of brain endothelial cells with astrocytes isolated from the same species. The novelty of the manuscript is very limited and the methodology is very basic. The significance of this manuscript could be improved by comparison of the described animal BBB models to at least one of the existing human BBB models.

6. PLOS authors have the option to publish the peer review history of their article (what does this mean?). If published, this will include your full peer review and any attached files.

Reviewer #1: No

Reviewer #2: No

---

## [Author Response · Author response to Decision Letter 0]

16 Sep 2020

Comments to the Author

1. Is the manuscript technically sound, and do the data support the conclusions?

Reviewer #1: Yes

Reviewer #2: Partly

2. Has the statistical analysis been performed appropriately and rigorously? 

Reviewer #1: Yes

Reviewer #2: Yes

3. Have the authors made all data underlying the findings in their manuscript fully available?

Reviewer #1: Yes

Reviewer #2: Yes

4. Is the manuscript presented in an intelligible fashion and written in standard English?

Reviewer #1: Yes

Reviewer #2: Yes

5. Review Comments to the Author

Reviewer #1: Thomsen and coauthors have evaluated property of the in vitro Blood brain barrier (BBB) generated from the three different species, mouse, rat and porcine. Analyzing the tight junction protein expression, brain endothelial barrier permeability (Papp and TEER assay) and expression of Pgp efflux transporter and transferrin receptor, they found significant differences in TEER values and brain endothelial barrier tightness between porcine comparing with rat and mouse BBB models, suggesting that all models have limitation and should be utilized based on the propose of study.

This is a very good study on an important topic. In general, the manuscript is well prepared, and the experiments are carefully designed and mapped out. Data is convincing and well-presented and discussed. The manuscript is very well written and statistical analysis is very rigorous. However, there are several concerns regarding this study, which should be addressed:

1. The authors should take in consideration that expression pattern of the tight junction (Tj)protein is different during the culturing/co-culturing and forming the barrier in vitro. It is not clear from presented data when was obtained samples, when the cells are confluent, or barrier is formed (5-7 days of culturing). It will be important to delineate the expression pattern of Tj mRNA expression in all three BBB models.

Our reply: We agree with the reviewer that this is important information to share, and accordingly we have specified when RNA was purified from the different BECs cultures in the material and methods section ”Gene expression analysis”.

2. Regarding the expression pattern of Tj proteins, it will be important to show the dynamic of protein expression together with mRNA.

Our reply: In this study, we focused on evaluating the protein expression using immunocytochemical stainings, instead of eg. Western blot analysis. Using the immunocytochemical staining technique we can evaluate the exact location of the tight junction (TJ) proteins rather than evaluating the amount of protein. We think the location of the TJ proteins is a more important factor to evaluate since the location of these is essential for maintaining the integrity of the BECs.

Concerning including a more dynamic protein expression analysis in the current manuscript, we think it would be most valuable if these protein samples were obtained, together with the RNA samples used for the gene expression analysis. Unfortunately, we do not have such samples, meaning that we would have to isolate and culture new primary cells from other animals and use these to set up new in vitro BBB models for isolation of protein samples. The results from these studies will therefore not be directly comparable to the gene expression analyses, already included in the manuscript. The immunocytochemical stainings were, however, obtained at the same timepoint as the RNA samples, which is the time point where we expected the cell cultures had reached the highest TEER values, hence making these two analyses comparable. 

3. Fig 3C is very important but it is very difficult to follow. Suggestion will be to split figure by species, keeping the same value on both x and y axis for easier comparison.

Our reply: We appreciate the reviewer's comments, however, we are not able to split the figure by species since we are directly comparing the cell size of the different species in figure 3C. Instead, we have tried to make the figure text and the results section addressing this figure more explanatory. 

4. The authors completely neglected the role of pericytes in the in vitro BBB coculture model. Pericytes were showed to play critical role for the barrier integrity/permeability. The current property could be only limited on astrocytes role. This should be addressed. 

Our reply: We agree with the reviewer that the significance of the pericytes are important, and have therefore also addressed this issue further in the introduction and discussion chapter. The present study aimed to compare the most widely used in vitro models of all three species, which is the main reason why we have not included pericytes in this study. We have further underlined this choice in both the introduction and discussion sessions. Pericytes are mainly known to regulate vesicular transport at the BBB and contribute to the maturation of tight junctions, and since we did not include pericytes in the present study we did not either include any analysis of vesicular transport. We do not believe that the addition of pericytes in the current analysis would add a great value to the study, especially since we have already previously published that the addition of pericytes did not significantly affect the tightness of the barrier (TEER, Papp) [1–3].

In addition to this, the inclusion of pericytes in the study at this timepoint would entail isolation of new primary cells from new sets of animals. All experiments would then have to be repeated using the triple culture model. In the present study, we have included cells from the same isolation batch to set up both mono and co-cultures, and these have always been analyzed side by side in each experiment to ensure comparable results. Adding the triple culture model based on new isolation batches at this time point and analyzing these individually, would not be directly comparable to the already included results. We, therefore, believe it will be of greater value to leave the influence of the pericytes for a future study, as also stated in the discussion chapter. 

5. One of the major goals of the study should be to compare these models to human in vitro BBB model. This will gain more translator character for the obtained results.

Our reply: We agree with the reviewer that human BBB models are important, and many laboratories use these models instead of the primary in vitro BBB models based on cells from animal sources. We have further addressed this in the discussion chapter. Additionally, a review further addressing the issue was recently published. In this review, they made a thorough comparison of different in vitro BBB models constructed from both cell lines and primary cells from human and animal sources. We have, therefore, included a direct reference to this review in our discussion chapter. Only the human BBB models based on stemcells can reach TEER values comparable to in vitro BBB models constructed from primary cells from animal sources. We, therefore, do not agree that including a human in vitro BBB should be a major goal of this study. Furthermore, we are not routinely using human in vitro BBB models in our laboratory, and would, therefore, need to establish one solely for this purpose. The strength of the current manuscript is, however, that the three in vitro BBB models that we have compared are very comparable because they are isolated based on the same protocol and widely used in our laboratory. None of this would apply to a human BBB model. Finally, several different human BBB models have been characterized in the literature, why including only one human BBB model, would probably not be of great value to give the manuscript a more translational character. 

Reviewer #2: The study compares barrier and enzymatic properties of monocultures of brain endothelial cells isolated from mouse, rat, and porcine brains as well as co-cultures of brain endothelial cells with astrocytes isolated from the same species. The novelty of the manuscript is very limited and the methodology is very basic. The significance of this manuscript could be improved by comparison of the described animal BBB models to at least one of the existing human BBB models.

Our reply: We accept and embrace the comments from the reviewer regarding the novelty and methodology of this study. We wanted to compare three in vitro BBB models using primary cells from animal sources to add important information to the ongoing discussion in both the literature and scientific community regarding choosing the right or best in vitro BBB model for future studies. To our knowledge, no studies have made a direct and thorough comparison of different models concerning this issue, although many have developed preconceptions about the best model, primarily based on reported TEER values. We, therefore, wanted to look further into this by making a comparison of the most widely used, and easily constructed in vitro BBB models from three different species. For the purpose we choose a methodology used and known by all laboratories working with in vitro BBB model, to make the manuscript relatable to most.

We, additionally, agree with the reviewer that human BBB models are important tools to study eg. BBB permeability of drugs intended for treating human disease. However, to this date, no standard human BBB model is widely used across laboratories worldwide. We are not routinely using human in vitro BBB models in our laboratory, and would, therefore, need to establish one solely for this purpose. The only human BBB models that can reach TEER values comparable to in vitro BBB models constructed from primary cells of animal sources are those based on stemcells. However, establishing such a model for inclusion in this manuscript is unrealistic as these models are known to be quite laborious. Including a human model based on immortalized cell lines would not be comparable in TEER or permeability to the models already included in the manuscript. The strength and the purpose of the current manuscript are, however, to compare the most widely used in vitro BBB models constructed from primary cells from animal sources isolated based on practically identical isolation protocol. All three models have previously been characterized in previous papers and widely used in our laboratory. None of this would apply to either of the human in vitro BBB models, as we would need to construct one solely for this purpose. Finally, several different human BBB models have been characterized in the literature, why including only one human BBB model, would probably not be of great value to give the manuscript a more translational character. We have already in the discussion chapter included a paragraph emphasizing the importance of developing human in vitro BBB models. In addition to this, a review has recently been published in which a thorough comparison of different in vitro BBB models constructed from both cell lines and primary cells from human and animal sources was performed. We have, therefore, included a direct reference to this review in our discussion chapter. We do therefore not agree that the novelty of this study is limited and that including a human in vitro BBB would significantly improve the impact of the manuscript.

References:

1. Burkhart A, Thomsen LB, Thomsen MS, Lichota J, Fazakas C, Krizbai I, et al. Transfection of brain capillary endothelial cells in primary culture with defined blood-brain barrier properties. Fluids Barriers CNS. 2015;12. doi:10.1186/s12987-015-0015-9

2. Thomsen MS, Birkelund S, Burkhart A, Stensballe A, Moos T. Synthesis and deposition of basement membrane proteins by primary brain capillary endothelial cells in a murine model of the blood–brain barrier. J Neurochem. 2017;140: 741–754. doi:10.1111/jnc.13747

3. Thomsen LB, Burkhart A, Moos T. A Triple Culture Model of the Blood-Brain Barrier Using Porcine Brain Endothelial cells, Astrocytes and Pericytes. PLoS One. 2015;10: e0134765. doi:10.1371/journal.pone.0134765 [doi]

---

## [Decision Letter · Decision Letter 1]

15 Oct 2020

PONE-D-20-21385R1

The blood-brain barrier studied in vitro across species

PLOS ONE

Dear Dr. Burkhart,

Thank you for submitting your manuscript to PLOS ONE. After careful consideration, we feel that it has merit but does not fully meet PLOS ONE’s publication criteria as it currently stands. Therefore, we invite you to submit a revised version of the manuscript that addresses the points raised during the review process.

ACADEMIC EDITOR: Please insert comments here and delete this placeholder text when finished. Be sure to:

Dear Dr. Burkhart

Despite the improvement the manuscript the critical question and suggestion to include human cells is not addressed. Please include those data and submit the manuscript. 

Eliseo

Please submit your revised manuscript by 60 days,If you will need more time than this to complete your revisions, please reply to this message or contact the journal office at plosone@plos.org. Please include the following items when submitting your revised manuscript:

We look forward to receiving your revised manuscript.

Kind regards,

Eliseo A Eugenin, Ph.D.

Academic Editor

PLOS ONE

Additional Editor Comments (if provided):

Dear Dr. Burkhart

Despite the improvement the manuscript the critical question and suggestion to include human cells is not addressed. Please include those data and submit the manuscript.

Eliseo

Reviewers' comments:

Reviewer's Responses to Questions

**Comments to the Author**

1. If the authors have adequately addressed your comments raised in a previous round of review and you feel that this manuscript is now acceptable for publication, you may indicate that here to bypass the “Comments to the Author” section, enter your conflict of interest statement in the “Confidential to Editor” section, and submit your "Accept" recommendation.

Reviewer #1: All comments have been addressed

Reviewer #2: (No Response)

2. Is the manuscript technically sound, and do the data support the conclusions?

Reviewer #1: Yes

Reviewer #2: Partly

3. Has the statistical analysis been performed appropriately and rigorously? 

Reviewer #1: Yes

Reviewer #2: Yes

4. Have the authors made all data underlying the findings in their manuscript fully available?

Reviewer #1: Yes

Reviewer #2: Yes

5. Is the manuscript presented in an intelligible fashion and written in standard English?

Reviewer #1: Yes

Reviewer #2: Yes

6. Review Comments to the Author

Reviewer #1: The authors respond to all arise questions/ concerns. There si not any further concerns regarding this study.

Reviewer #2: The authors did not satisfactorily addressed my concerns. The major concerns remain 1) limited novelty, 2) limited significance, and 3) very basic methodology. Without comparison to a human BBB model this study has very limited importance. Not including pericytes in the models is another major limitation of the experimental design. This study does not advance the BBB field in any significant manner.

7. PLOS authors have the option to publish the peer review history of their article (what does this mean?). If published, this will include your full peer review and any attached files.

Reviewer #1: No

Reviewer #2: No

---

## [Author Response · Author response to Decision Letter 1]

21 Dec 2020

Comments to the Author

1. If the authors have adequately addressed your comments raised in a previous round of review and you feel that this manuscript is now acceptable for publication, you may indicate that here to bypass the “Comments to the Author” section, enter your conflict of interest statement in the “Confidential to Editor” section, and submit your "Accept" recommendation.

Reviewer #1: All comments have been addressed

Reviewer #2: (No Response)

2. Is the manuscript technically sound, and do the data support the conclusions?

Reviewer #1: Yes

Reviewer #2: Partly

3. Has the statistical analysis been performed appropriately and rigorously? 

Reviewer #1: Yes

Reviewer #2: Yes

4. Have the authors made all data underlying the findings in their manuscript fully available?

Reviewer #1: Yes

Reviewer #2: Yes

5. Is the manuscript presented in an intelligible fashion and written in standard English?

Reviewer #1: Yes

Reviewer #2: Yes

6. Review Comments to the Author

Reviewer #1: The authors respond to all arise questions/ concerns. There are not any further concerns regarding this study.

Our reply: We are pleased that the reviewer is satisfied with the revision of the manuscript and that we have responded fully to the questions/concerns raised. 

Reviewer #2: The authors did not satisfactorily address my concerns. The major concerns remain 1) limited novelty, 2) limited significance, and 3) very basic methodology. Without comparison to a human BBB model, this study has very limited importance. Not including pericytes in the models is another major limitation of the experimental design. This study does not advance the BBB field in any significant manner.

Our reply: We appreciate the comments from the reviewer and have revised the manuscript accordingly to the raised concerns. Throughout the manuscript, we have addressed the issue of lack of novelty by emphasizing the significance of this study in regards to adding value and advantage to the field of BBB research. We have emphasized the main message of the manuscript and included a chapter in our discussion where we thoroughly compare our animal in vitro BBB models to previously published human in vitro BBB models to add a more translational perspective to the manuscript. Furthermore, we have included a scientific discussion on the use of the different models. We are, unfortunately, not able to include the role of the pericytes, for reasons augmented in the previous revision. We agree that the role of the pericytes is important, and will address this in a future study.

---

## [Decision Letter · Decision Letter 2]

1 Feb 2021

PONE-D-20-21385R2

The blood-brain barrier studied in vitro across species

PLOS ONE

Dear Dr. Burkhart,

Thank you for submitting your manuscript to PLOS ONE. After careful consideration, we feel that it has merit but does not fully meet PLOS ONE’s publication criteria as it currently stands. Therefore, we invite you to submit a revised version of the manuscript that addresses the points raised during the review process.

ACADEMIC EDITOR:

Dear Dr. Burkhart: My apologies for the long time to take a decision in this manuscript. Please correct the minor comments suggested by the reviewers 

We look forward to receiving your revised manuscript.

Kind regards,

Eliseo A Eugenin, Ph.D.

Academic Editor

PLOS ONE

Additional Editor Comments (if provided):

Dear Dr. Burkhart:

My apologies for the long time to take a decision in this manuscript. Please correct the minor comments suggested by the reviewers

Eliseo Eugenin

Reviewers' comments:

Reviewer's Responses to Questions

**Comments to the Author**

1. If the authors have adequately addressed your comments raised in a previous round of review and you feel that this manuscript is now acceptable for publication, you may indicate that here to bypass the “Comments to the Author” section, enter your conflict of interest statement in the “Confidential to Editor” section, and submit your "Accept" recommendation.

Reviewer #1: All comments have been addressed

Reviewer #3: All comments have been addressed

Reviewer #4: All comments have been addressed

2. Is the manuscript technically sound, and do the data support the conclusions?

Reviewer #1: Yes

Reviewer #3: Yes

Reviewer #4: Yes

3. Has the statistical analysis been performed appropriately and rigorously? 

Reviewer #1: Yes

Reviewer #3: Yes

Reviewer #4: N/A

4. Have the authors made all data underlying the findings in their manuscript fully available?

Reviewer #1: Yes

Reviewer #3: Yes

Reviewer #4: Yes

5. Is the manuscript presented in an intelligible fashion and written in standard English?

Reviewer #1: Yes

Reviewer #3: Yes

Reviewer #4: Yes

6. Review Comments to the Author

Reviewer #1: All concerns are addressed satisfactorily . The manuscript is significantly improved. I do nbot ahve any concern regarding this study.

Reviewer #3: This is a very good contribution to the BBB study, a lot of effort from authors. I observed only grammatical errors such proper use of punctuation is suggested,

In line 265, " Samples

266 from the opposite chamber from where [3H]-digoxin" Please write in detail what is your sample, like cell culture media.

Also you mention most of the places mono culture and co-culture, you must define in sentence co-cultre with astrocytes or something else, it will be easy for the reader to follow your statement.

Line 281: the mean cell area , is it from a single cell or a single junction, justify why you have only n =4

line 516: again mention co-culture with astrocytes !

Figure 1(B), you must add scale bar in each image

Figure 1(C), you added scale bar to only one, you must add to all

Figure 2 (B), add scale bar to all, also coculture (must write culture name)

similarly , figure 4 and 5, add scale bar.

Figure 4 (C), co-culture of mBEC's , cell are rounded up , if you observe any toxicity, you must report in the manuscript.

Reviewer #4: In their study, Burkhart & al. reviewed the transendothelial electric resistance (TEER) of 3 different animal model of blood brain barrier (BBB): porcine, mice and rat. They also analyzed the difference in permeability using [3H]-D Mannitol. They found a greater TEER in the porcine model but with comparable permeability. They also analyzed the expression pattern of tight junction. They conclude that having a higher TEER in BBB model might not be of interest since permeability stays the same and they suspect that the bigger cell size is the origin of the increase in TEER.

The authors have well defined the existing problematic: which in vitro BBB model is the closest to the in vivo model. They thoroughly explain and justify why their model do not use pericytes, even though this would be of interest in futures studies as those models would be closer to in vivo situation and detail the advantages and limitations of every model.

Yet, as the authors state, the TEER value of different models as already been studied and reported in the literature. The most interesting results is the analysis of permeability and this could be more emphasized throughout the manuscript. The study of the expression of tight junction is interesting but I would suspect a model with pericytes to come closer to the reality.

I have some minor comments. The structure of the article is not well suited. Parts of results and conclusion are already given in the introduction as well as pieces of discussion. Legends of figures should be written under the figures and not in the text. The text could be reviewed for a few typos.

7. PLOS authors have the option to publish the peer review history of their article (what does this mean?). If published, this will include your full peer review and any attached files.

Reviewer #1: No

Reviewer #3: **Yes: **Shahnaz Majid Qadri

Reviewer #4: No

---

## [Author Response · Author response to Decision Letter 2]

23 Feb 2021

Response to the reviewers

1. If the authors have adequately addressed your comments raised in a previous round of review and you feel that this manuscript is now acceptable for publication, you may indicate that here to bypass the “Comments to the Author” section, enter your conflict of interest statement in the “Confidential to Editor” section, and submit your "Accept" recommendation.

Reviewer #1: All comments have been addressed

Reviewer #3: All comments have been addressed

Reviewer #4: All comments have been addressed

2. Is the manuscript technically sound, and do the data support the conclusions?

Reviewer #1: Yes

Reviewer #3: Yes

Reviewer #4: Yes

3. Has the statistical analysis been performed appropriately and rigorously? 

Reviewer #1: Yes

Reviewer #3: Yes

Reviewer #4: N/A

4. Have the authors made all data underlying the findings in their manuscript fully available?

Reviewer #1: Yes

Reviewer #3: Yes

Reviewer #4: Yes

5. Is the manuscript presented in an intelligible fashion and written in standard English?

Reviewer #1: Yes

Reviewer #3: Yes

Reviewer #4: Yes

6. Review Comments to the Author

Reviewer #1: All concerns are addressed satisfactorily. The manuscript is significantly improved. I do not have any concerns regarding this study.

Reviewer #3: This is a very good contribution to the BBB study, a lot of effort from authors. I observed only grammatical errors such as proper use of punctuation is suggested.

Our reply: We have carefully re-read the manuscript to remove grammatical errors.

In line 265, " Samples from the opposite chamber from where [3H]-digoxin" Please write in detail what is your sample, like cell culture media.

Our reply: We have rephrased the sentence according to the reviewer's suggestion. 

Also you mention most of the places mono culture and co-culture, you must define in sentence co-culture with astrocytes or something else, it will be easy for the reader to follow your statement.

Our reply: We appreciate the comments and agree with the reviewer. We have therefore clearly defined the use of co-culture in the method section, when this definition is used for the first time, and have furthermore specified this throughout the manuscript.

Line 281: the mean cell area, is it from a single cell or a single junction, justify why you have only n =4

Our reply: We have rephrased this section to specify exactly how this is calculated. We have calculated the average cell area per image and used the mean cell area of four different images taken from two separate hanging culture inserts in the analysis. The total n value of four is therefore a mean calculated on the basis of several cells. The n=4 is furthermore from each culturing condition, meaning a total of eight images were analyzed per species, and, as can also be seen in figure 3, the difference between mono and co-culture within each species is not highly variable, why we believe that including more images will not change the conclusion. We furthermore believe that it is more correct to use the average cell area per image, than entering the size of all single cells marked in total. 

line 516: again mention co-culture with astrocytes!

Our reply: We have specified this in the text. 

Figure 1(B), you must add scale bar in each image

Figure 1(C), you added scale bar to only one, you must add to all

Figure 2 (B), add scale bar to all, also coculture (must write culture name)

similarly, figure 4 and 5, add scale bar.

Our reply: We have added scale bars on all images in all five figures. 

Figure 4 (C), co-culture of mBEC's, cell are rounded up, if you observe any toxicity, you must report in the manuscript.

Our reply: We understand that it might look somehow like that, however, this is not the case. We have carefully inspected the cells using only the nuclear stain, and don’t observe any toxicity or rounding up of the cells. It might simply be the P-gp expression pattern, that is causing it to look like this. The cells used for these stainings have not been subjected to anything that might cause toxicity. They have simply been cultured to confluency, received tight junction inducing agents, and fixated at the time of high TEER, like all the other cells used for immunocytochemical stainings. 

Reviewer #4: In their study, Burkhart & al. reviewed the transendothelial electric resistance (TEER) of 3 different animal models of blood-brain barrier (BBB): porcine, mice, and rat. They also analyzed the difference in permeability using [3H]-D Mannitol. They found a greater TEER in the porcine model but with comparable permeability. They also analyzed the expression pattern of tight junction. They conclude that having a higher TEER in BBB model might not be of interest since permeability stays the same and they suspect that the bigger cell size is the origin of the increase in TEER.

The authors have well defined the existing problematic: which in vitro BBB model is the closest to the in vivo model. They thoroughly explain and justify why their model does not use pericytes, even though this would be of interest in futures studies as those models would be closer to in vivo situation and detail the advantages and limitations of every model.

Yet, as the authors state, the TEER value of different models has already been studied and reported in the literature. The most interesting results is the analysis of permeability and this could be more emphasized throughout the manuscript. The study of the expression of tight junction is interesting but I would suspect a model with pericytes to come closer to the reality.

Our reply: We have revised the manuscript to add more emphasis on the main finding of this study. We agree that the role of the pericytes is highly relevant in this content and likewise important, why we plan to address this in a future study.

I have some minor comments. The structure of the article is not well suited. Parts of the results and conclusion are already given in the introduction as well as pieces of discussion. 

Our reply: We appreciate the comment and have revised the structure of the introduction, which no longer includes results, discussion, or conclusions

Legends of figures should be written under the figures and not in the text. 

Our reply: The legends are included in the text at first mentioning due to formatting criteria given in the “instruction to the authors” provided by PLOS One 

The text could be reviewed for a few typos.

Our reply: We appreciate the comment and have carefully re-read the manuscript to remove typos.

---

## [Editor Report · Decision Letter 3]

25 Feb 2021

The blood-brain barrier studied in vitro across species

PONE-D-20-21385R3

Dear Dr. Burkhat,

We’re pleased to inform you that your manuscript has been judged scientifically suitable for publication and will be formally accepted for publication once it meets all outstanding technical requirements.

Kind regards,

Eliseo A Eugenin, Ph.D.

Academic Editor

PLOS ONE

Additional Editor Comments (optional):

Dear Dr. Burkhart

Thank you for performing all the changes suggested and my apologies for the extended time for the review process

Best Regards

Eliseo
---

## [Editor Report · Acceptance letter]

3 Mar 2021

PONE-D-20-21385R3 

The blood-brain barrier studied *in vitro* across species  

Dear Dr. Burkhart:

I'm pleased to inform you that your manuscript has been deemed suitable for publication in PLOS ONE. Congratulations! Your manuscript is now with our production department. 

Kind regards, 

on behalf of

Dr. Eliseo A Eugenin 

Academic Editor

PLOS ONE